# Fatty acid binding protein 1 and fatty acid synthetase over-expression have differential effects on collagen III synthesis and cross-linking in Zongdihua pig primary adipocytes

**Rong Yang[1], Di Zhou[1]\*, Zhihong Yan[2], Zhonghai Zhao[3], Yan Wang[1], Jun Li[1], Liqun Ren[1], Lingling Xie[1], Xin Wang[4]**

**1** Guizhou Provincial Breeding Livestock and Poultry Germplasm Determination Center, Guiyang, Guizhou, China, **2** College of Animal Science, Guizhou University, Guiyang, Guizhou, China, **3** Zunyi Animal Husbandry and Fishery Station, Zunyi, Guizhou, China, **4** Guizhou Animal Husbandry and Veterinary Research Institute, Guiyang, Guizhou, China

\* 250161818@qq.com

**Data Availability Statement:** All relevant data are within the manuscript and its Supporting Information files.

## Abstract

The purpose of this study was to determine whether FABP1 and FAS regulate expression of collagen and its crosslinking via lysyl oxidase in isolated adipocytes from Zongdihua pigs. We aimed to identify biochemical processes affecting meat quality using molecular tools to provide a basis for breeding improvement of these animals. We measured expression levels of FABP1 and related genes using qRT-PCR in *longissimus dorsi* muscle and subcutaneous adipose tissues. Primary adipocytes from fat tissues were isolated and FABP1 and FAS were over-expressed from recombinant plasmids. Sequence analysis of the cloned genes indicated that FABP1 encodes a hydrophobic protein of 128 amino acids and contained 12 predicted phosphorylation sites and no transmembrane regions. The basal levels of FABP1 and FAS expression in pig tissues were 3–3.5-fold higher in subcutaneous fat compared with muscle ($P < 0.01$). Recombinant expression plasmids were successfully transfected into the cloned preadipocytes and (a) over-expression of FAS resulted in significantly increased expression of COL3A1 ($P < 0.05$) and significantly inhibited lysyl oxidase *LOX* expression ($P < 0.01$); (b) over-expression of FABP1 significantly increased COL3A1 expression ($P < 0.01$) and significantly inhibited *LOX* expression ($P < 0.05$) and significantly reduced lysyl oxidase activity ($P < 0.01$). Therefore, FAS enhanced FABP1 expression resulting in increased collagen accumulation and this preliminarily suggested that FAS and FABP1 can serve as fat-related candidate genes and provide a theoretical basis for the study of fat deposition in Zongdihua pigs.

## 1 Introduction

The extracellular matrix (ECM) play roles in adipocyte proliferation, differentiation, and migration. Moreover, ECM anchors these cells to prevent mechanical movement and protect

**Funding:** This study was supported in part by two funding sources: 1) A special project for central guided local development funded from the Science and Technology Department of Guizhou Province to Dr. D. Zhou (Project No.: Qianke Central Land [2018]4015). Co-participants including R. Yang, Z. Zhao, Y. Wang, J. Li, L. Ren and, L. Xie. 2) A joint research project of local poultry industry in Guizhou province funded by Guizhou Provincial Department of Finance to Dr. D. Zhou (Project Number: Qian Cai Nong [2020] No. 175). The funders had no role in study design, data collection and analysis, decision to publish, or preparation of the manuscript.

**Competing interests:** The authors have declared that no competing interests exist.

mature adipocytes from external compression and stimulation. It is well established that collagen is the main component of ECM [1, 2]. Large quantities of collagen are synthesized in the process of fat deposition [3]. These collagen-encoding genes are also highly expressed in tissues with higher fat contents and transforming growth factor β (TGF-β) is a positive regulator of these genes [4, 5]. The metalloproteinase ADAMTS2 cleaves the N-prodomains of these fibrillar collagens and the expression of this gene is significantly positively correlated with intramuscular fat formation. In general, there is an intrinsic link between collagen and fat biosynthesis but the specific regulatory mechanisms are unclear.

The primary component of the ECM is Type III collagen encoded by COL3A1 that is extensively synthesized during fat deposition [3]. Lysyl oxidase (LOX) catalyzes collagen and elastin crosslinking and adds to the tensile strength of the ECM [5]. The ECM also serves as a storage compartment for components involved in protein-protein interactions such as TGF-β, which is a positive regulator of collagen genes and it is highly expressed in tissues with high fat content [6]. In particular, fatty acid binding protein (FABP) family members such as FABP1 play key roles in fat deposition as well as synthesis and degradation of body fat [7–9]. The anabolic enzyme fatty acid synthase (FAS) is also highly expressed in adipocytes and catalyzes triglyceride synthesis and thereby is a regulator of fat deposition [10, 11].

The FABP1 and FAPB2 genes play important roles in the regulation of fat metabolism and FABP1 gene is expressed in numerous tissues in pigs [12]. A previous study indicated that duck L-FABP or FAPB1 gene polymorphisms were significantly correlated with breast muscle fat content and expressed in other tissues [13]. Thus, it was speculated that the L-FABP gene may be the primary gene regulating pig intramuscular fat content or is a closely linked marker gene [14]. In this study, expression levels of the FABP1 and FAS genes in *longissimus dorsi* and subcutaneous adipose tissues of Zongdihua pigs were measured with qPCR technology and vectors expressing FAS and FABP1 were constructed to examine the roles of these genes in fat deposition.

## 2 Materials and methods

### 2.1 Ethics statement

This study was carried out in strict accordance with the guidelines for the use of laboratory animals in Guizhou University. This protocol was approved by the Experimental Animal Ethics Review Committee of Guizhou University (protocol number: EAE-GZU-2021-P019). All operations were performed under anesthesia with sodium pentobarbital and every effort was made to minimize pain.

### 2.2 Test samples

Healthy Zongdihua pigs including 6 each 9-month and 3-day old pigs with similar body weights for each group were obtained from Ziyun County Ziwei Animal Husbandry. Samples (2 g) of subcutaneous fat and *longissimus dorsi* muscle were collected and placed in an RNA preservation solution (Sartoris. Kibbutz Beit-Harmek, Israel) and then frozen in liquid nitrogen and stored at -80˚C for further analyses.

### 2.3 RNA extraction, *mRNA* expression analysis and qRT-PCR

Total RNA was extracted from 9 months old Zongdihua pigs homogenized fat and *longissimus dorsi* muscle using Trizol with the protocol recommended by the manufacturer (Invitrogen, Carlsbad, CA, USA). First strand cDNA was synthesized using the HiFi Script cDNA kit as recommended by the manufacturer (Thermo Fisher). The concentration and purity of RNA and

cDNA was determined using a micro-UV spectrophotometer (Nanodrop, Thermo Fisher). Real-time quantitative PCR (qRT-PCR) was used to measure steady state *mRNA* levels from tissues and cells with GAPDH as the internal reference gene and FABP1 and FAS as the quantitative targets. The reactions (10 µL) in triplicate contained 5 µL SsoFASt EvaGreen Supermix, 5 pmol each primer, and 1 µL cDNA template that were cycled at 95˚C for 3 min and 39 cycles of 95˚C for 15 s, 60˚C for 15 s and 72˚C for 1 min and collection time for 5 sec. Melting curve analysis to determine amplicon purity was performed at increments of 0.5˚C from 60 to 95˚C (S5 Fig). The 2-$^{\triangle\triangle}$Ct method was used to process and analyze the data [15–17].

Transfected cells were used to assess the effects of the over-expression of FABP1 and FAS on COL3A1 and LOX gene expression. Cells were plated in 6-well plates, allowed to grown to 90% confluence and transfected. The cells were removed after 36 h at 37˚C and steady state *mRNA* was measured using qRT-PCR.

## 2.4 Construction of FABP1 and FAS plasmid expression vectors

Subcutaneous fat cDNA from the 3-day old pigs was used as template to PCR-amplify the CDS regions of the FABP1 and FAS genes. Amplicons were generated using the 2× Ex Taq kit (Thermo) with the following conditions: 95˚C for 5 min and 35 cycles of 95˚C for 30 s, 60˚C for 45 s and 72˚C for 30 s followed by a final extension at 72˚C for 5 min. The PCR products were recovered from agarose gels using a commercial kit (Omega Bio-Tek, Norcross, GA, USA) and cloned using the TA vector pUCM-T vector (Bio Basic, Ontario, Canada) and transformed into TOP 10 competent cells (Invitrogen) using the manufacturers specifications. Cloned versions of FABP1 and FAS were constructed and fused to the eGFP protein at their carboxyl termini and over-expressed via the strong CMV promoter. The transformation mixes were plated on ampicillin agar plates and incubated at 37˚C overnight. Single colonies were incubated in Luria Bertani broth containing ampicillin at 37˚C for 8 h. Possession of cloned genes were verified by PCR and further sequenced by Qingke Biological to confirm insert identity. The cloned inserts were released by PstI and KpnI digestion and cloned into the eukaryotic expression vector pEGFP-C1 (Clontech, Takara Bio, Kyoto, Japan) using the above procedure but with kan selection and plasmid insert identities were verified by DNA sequencing. A comparison of the sequencing results in NCBI showed that the results were consistent with the original sequence. The physicochemical properties, secondary structures, transmembrane regions, and phosphorylation sites of FABP1 and FAS proteins were analyzed by the bioinformatics analysis software.

## 2.5 Isolation and culture of adipocytes

Healthy 3-day-old Zongdihua pigs were used for isolation of adipose cells. Subcutaneous adipose tissues were collected and quickly washed in PBS lacking $Ca^{2+}$ and $Mg^{2+}$ and connective tissues and blood vessels were removed. The tissue was then cut into pieces and digested with type II collagenase for 90 min. The digestion was terminated by adding DMEM/F 12 medium containing 10% serum and the mixture was filtered through sterile gauze and a cell sieve (200-mesh and 400-mesh). Cells from the filtrate were collected by centrifugation at 1500 rpm for 10 min and suspended in serum-free medium, mixed by pipetting and then centrifuged. The fat cell of 3-day-old pigs were cultured and the pellets were suspended in 10% DMEM / F12 complete medium and transferred to a cell culture flask and placed at 37˚C in a 5% $CO_2$ atmosphere for 4 h. The medium was replaced and the cells were incubated. When the cells reached approximately 85% confluence the cells were removed using trypsin / EDTA and transferred to 6-well plates and grown to 90% confluence. The medium was replaced and culture was continued for a total of 2–3 days. Differentiation was induced by replacing the growth

media with induction medium (10% DMEM/F12 complete medium, DEX stock solution, IBMX concentrated stock solution and insulin stock solution) and changed every 2 days to induce differentiation. The cells were observed for the production of lipid droplets using an inverted microscope. When lipid droplets were present in about 80% of the cells, Oil Red O staining was performed to identify adipocytes. The amount of stain present in the wells was quantified using spectrometry at 510 nm in a microplate reader.

## 2.6 Transfection of cultured adipocytes

Cultures identified as adipocytes were transfected at 80% confluence by replacing the maintenance medium with the following components: 4 μg recombinant plasmid DNA from pEGFP-C1-FABP1 and pEGFP-C1-FAS separately were combined with 125 μL OPTI-MEM and 5 μL P3000. Separately, in 5 μL Lipofectamine 2000 125 μL OPTI-MEM were combined. The mixtures were then combined and mixed and placed at 37°C in a 5% $CO_2$ atmosphere for 15 min. Transfected cells were inoculated in 6-well plates that was cultures for 24 h and examined for LOX activity.

The culture medium from transfected cells were used to assay for LOX enzyme activity using a commercial kit (Jianglai Biology, Beijing, China) and activity was measured as increased absorbance at 450 nm using a microplate reader. LOX activity in the sample was calculated according to a standard curve constructed using Excel.

## 2.7 Data processing and analysis

The data from all experiments were first tested for collation and outliers. Statistical analysis was performed using SPSS software, version 20.0 (IBM-SPSS, Chicago, Illinois, USA).One-way ANOVA and Duncan multiple comparisons test were used to evaluate the differences expression of different genes in different varieties. The data were presented as mean±standard deviation (SD). P < 0.01 indicated very significant differences and P < 0.05 significant differences. DNAStar (Madison, WI, USA) was used for DNA sequence analysis.

# 3 Results

## 3.1 Differential expression analysis

Our preliminary experiments examined the expression of FABP1, FAS, LOX and COL3A1 in subcutaneous fat and *longissimus dorsi* muscle tissues of the Zongdihua pigs. Primer design and synthesis using Primer 5 software (http://www.premierbiosoft.com/primerdesign/) was used to design primers for the porcine GAPDH, FABP1, FAS, LOX and COL3A1. GAPDH served as internal reference gene. The accession genes were downloaded from GenBank (https://www.ncbi.nlm.nih.gov/genbank/). All primers were synthesized by Qingke Biotechnology (Table 1).

The expression levels for all 4 genes were significantly (P < 0.01) elevated in subcutaneous fat tissues compared with the muscle (control) tissues (Fig 1). We therefore generated PCR amplicons for FABP1 and FAS to be used for cloning and over-expression analysis. We successfully amplified FAS and FABP1 (S1 Fig) that were then cloned using the TA vector pUCM-T(PCR TA Cloning Kit, Thermo Scientific) and transferred to the pEGFP-C1 expression vector. The predicted FAS gene encoded a protein of 333 amino acids (37.593 kDa) with pI = 7.377, pH 7.0. The FABP1 gene product was composed of 128 amino acids (14.107 kDa) and pI = 7.094 (Fig 2). FAS contained 22 potential S/T phosphorylation sites and 4 potential tyrosine phosphorylation sites while FABP1 possessed 12 S/T phosphorylation sites (Fig 3).

**Table 1. PCR primers used in this study.**

| Gene | Numbers | Sequence (5'→3') | Annealing temperature/°C | Product length/bp | Purpose |
|------|---------|------------------|--------------------------|-------------------|---------|
| *FABP1* | NM_001004046.2 | F:TTGGGAGAGGAGTGTGAGATG R:CACAGACTTGATGCCTTTGAA | 58 | 111 | qPCR |
| *FAS* | NM_213839.1 | F:CCTGTATCGCTGGACCACT R:GGGCACTCAGACTCCCTTT | 57.9 | 103 | qPCR |
| *GAPDH* | XM_021091114.1 | F:TTGTGATGGGCGTGAACC R:GTCTTCTGGGTGGCAGTGAT | 60.0 | 169 | qPCR |
| *LOX* | NM_001206403 | F:TACCAAGCCGACCAAGATA ' R:GTGTGCAGTACAGGCAAAT | 57.2 | 147 | qPCR |
| *COL3A1* | NM_001243297 | F:TTTGCTCTACTTCATCCCACT R:CTTCCAGACATCTCTATCCGC | 56.5 | 105 | qPCR |
| *FABP1*-CDS | NM_001004046.2 | F: CCGATGAACTTCTCCGGCAAATACC R:CACGGTACCCTCTTGCTGATTCTCTTGAAGACAA | 58.5 | 384 | Cloning |
| *FAS*-CDS | NM_213839.1 | F:CCGATGTCCGGGATCTGGGTTCT R:CACGGTACCCTAGGTCAAGCTTTCATTCTCATTC | 60.0 | 999 | Cloning |

Shaded areas in primer sequences of FABP1-CDS and FAS-CDS denote added protecting bases; Restriction endonuclease sites used for cloning are underlined (Table 1).

Additional analyses of predicted structures of the FAS and FABP1 proteins indicated they were typical for these proteins in other pig breeds.

## 3.2 Adipocyte culture and analysis

We utilized the subcutaneous fat of Zongdihua pigs to isolate pure cultures of adipocytes using differential adhesion. The subcutaneous adipocytes appeared round and adhered to the cell

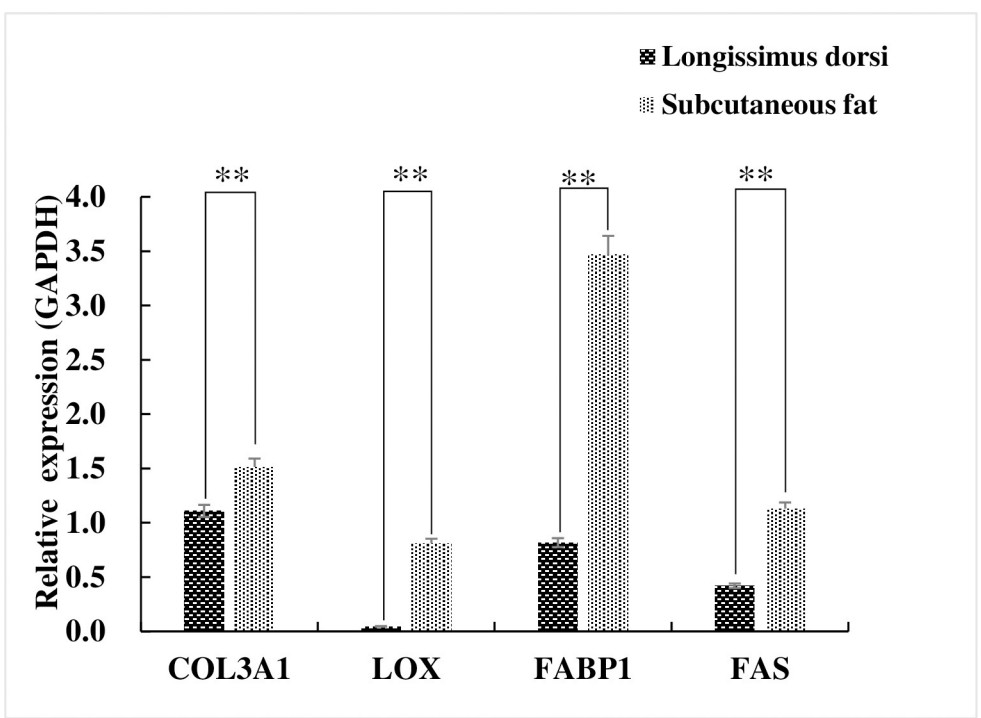

**Fig 1. Fat deposition and relative expression of collagen-related genes in Zongdihua pig adipose and muscle tissues.** **, p<0.01.

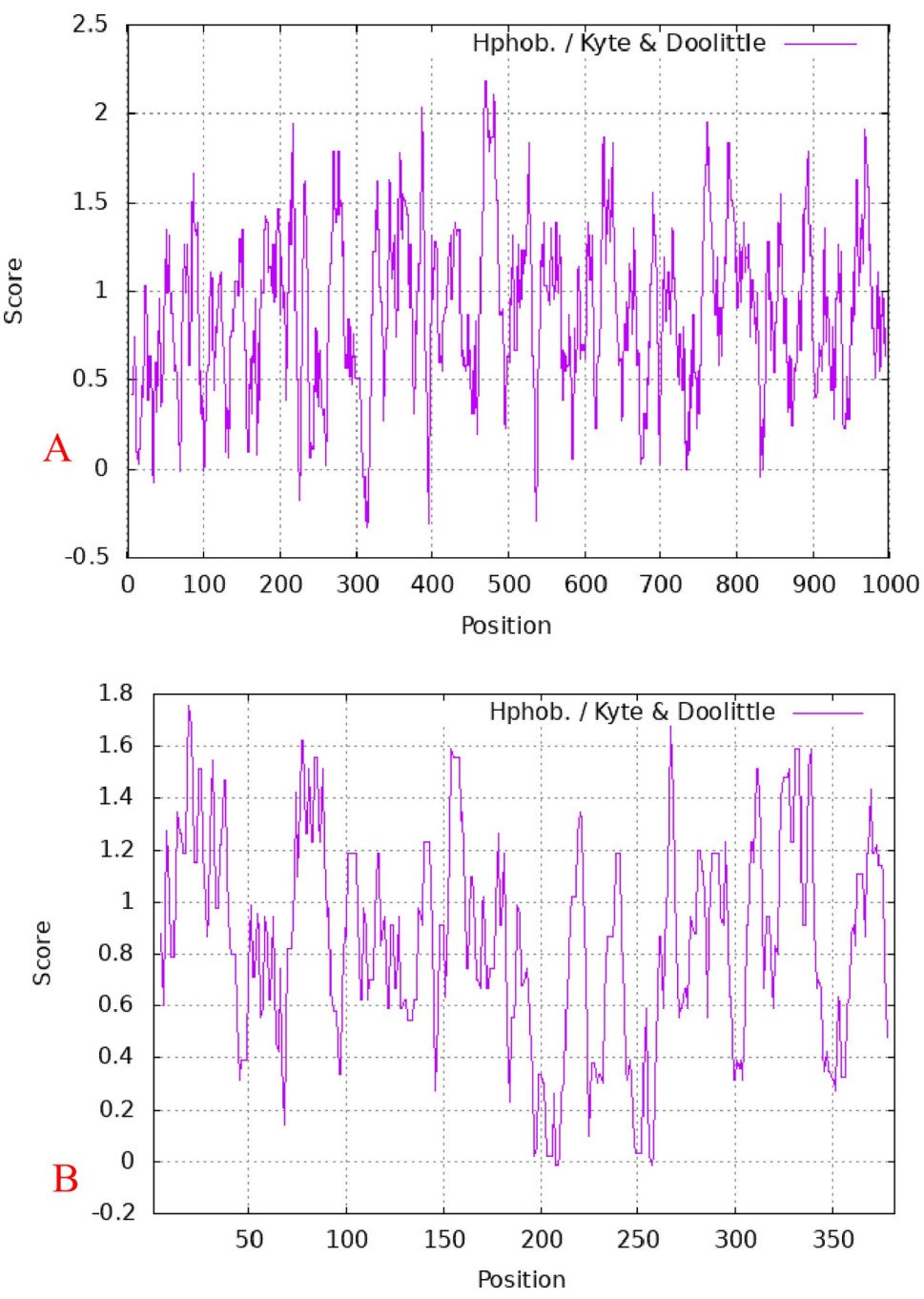

**Fig 2. Affinity/hydrophobicity analysis of FAS and FABP1 proteins from Zongdihua pigs.** (A) FAS (B) FABP1.

culture flask after 4 h in culture and were 80% confluent after 24 h. A subset of the cells appeared as long spindles with transparent bodies (S2A and S2B Fig). After 48 h in culture, the cells had proliferated and most of the cells appeared as long spindles and these reached confluence by 72 h (S2C and S2D Fig).

The identity of these cells as adipocyte precursors could be confirmed if they were able to differentiate in the presence of differentiation induction medium and be stained with Oil red O(Solarbio Biotech Company,Beijing, China). We found that the long spindles of the

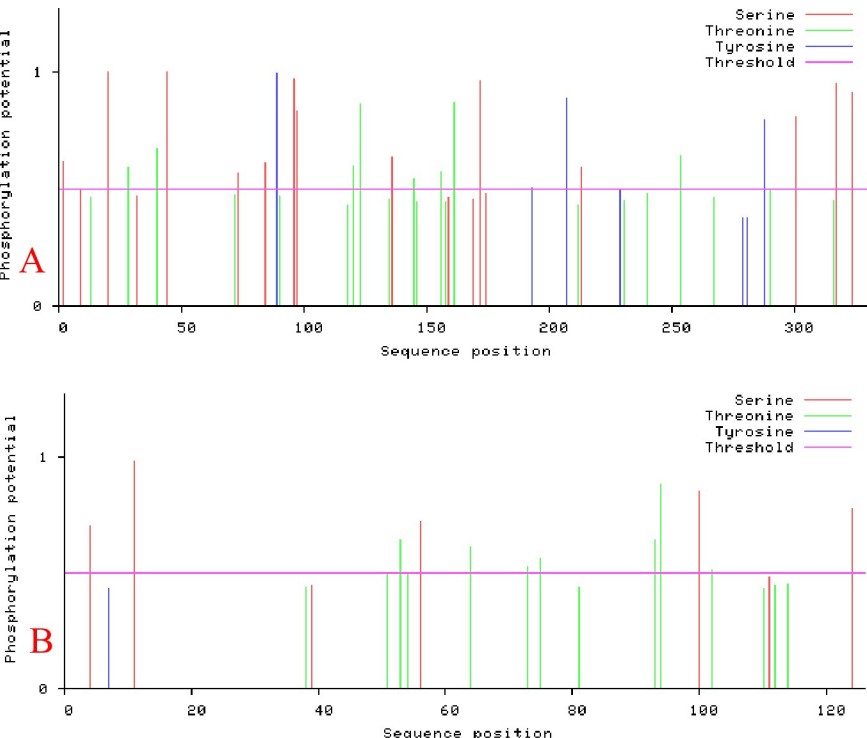

Note: The abscissa represents the number of amino acids, the ordinate represents the score of phosphorylation sites, Threshold is the dividing line (0.5), above the dividing line, it indicates phosphorylation sites, A: *FAS*; B: *FABP*1

**Fig 3. Analysis of Zongdihua pig FAS and FABP1 protein phosphorylation sites.** (A) FAS (B) FABP1.

precursor cells gradually became elliptical following 24 h exposure to the differentiation medium (S3A Fig). Lipid droplets began to appear in the cells 72–96 h (S3B Fig). After 6 days of induction and culture, the number of lipid droplets increased significantly and the volume increased (S3C Fig). The lipid droplets fused with the cells after 8 d. At this time, differentiation maintenance medium was replaced with induction medium and the cultures were allowed to continue for another 4–5 d. The round and transparent bodies in the cells were verified as lipid droplets by oil red O staining (S3D Fig). This confirmed that our primary cultures were differentiated into adipocytes.

### 3.3 Transient over-expression analysis of FAS and FABP1 in differentiated adipocytes

The transfection procedure for the cells was followed using fluorescent microscopy and 24h following transfection with Lipofectamine 3000 (Invitrogen, Carlsbad, CA, USA), clear and bright green protein expression was seen for both plasmid constructs compared with cells transfected with the vector only or with a blank control (S4A–S4E Fig). This indicated that the eGFP fusions were successfully expressed in the cells. These cells were then examined for the ability of the cloned genes to stimulate expression of COL3A1 and LOX using real-time quantitative PCR. We found that 36 h following transfection, *mRNA* levels for both FAS and FABP1 were 3–3.5-fold higher in the transfected cells versus controls (P<0.01). Interestingly, over-expression of the cloned genes resulted in significantly higher levels for COL3A1 (FAS gene, P<0.05; FABP1 gene, P<0.01) and lower levels for LOX (P<0.01;Fig 4).

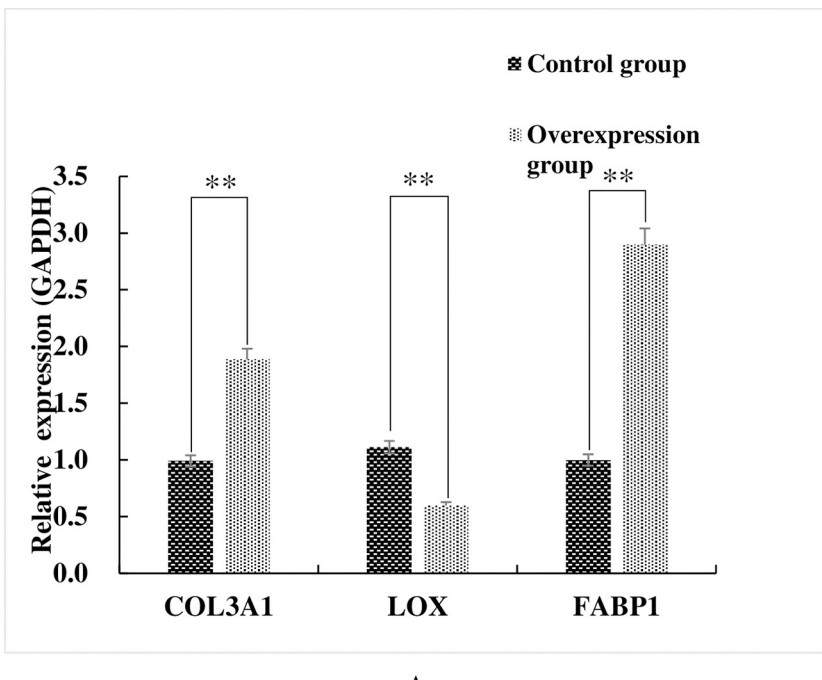

A

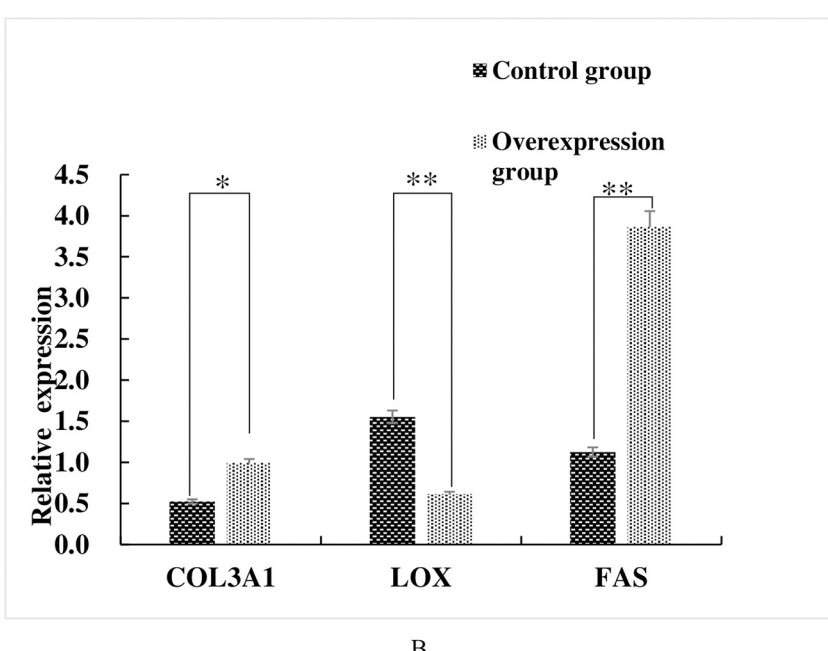

B

**Fig 4. Over-expression of FAS and FABP1 stimulate Col3A1 and inhibit LOX expression.** Cultured adipocytes were transiently transfected with (A) pEGFP-C1-FABP1 and (B) pEGFP-C1-FAS. Steady state *mRNA* levels for the indicated genes were determined using real time qRT-PCR. *, p < 0.05; **, *p < 0.01*.

We further examined the effects of transfected plasmids on LOX *mRNA* levels and lysyl oxi-dase activity. The supernatants of the transfected cells used above were examined for lysyl oxi-dase activity using a double-antibody sandwich ELISA method. LOX activity was measured using a commercial kit using the manufacturers protocol (Shanghai Jianglai Biological). We generated a linear standard curve for the assay (Fig 5B). Interestingly, the lysyl oxidase activity

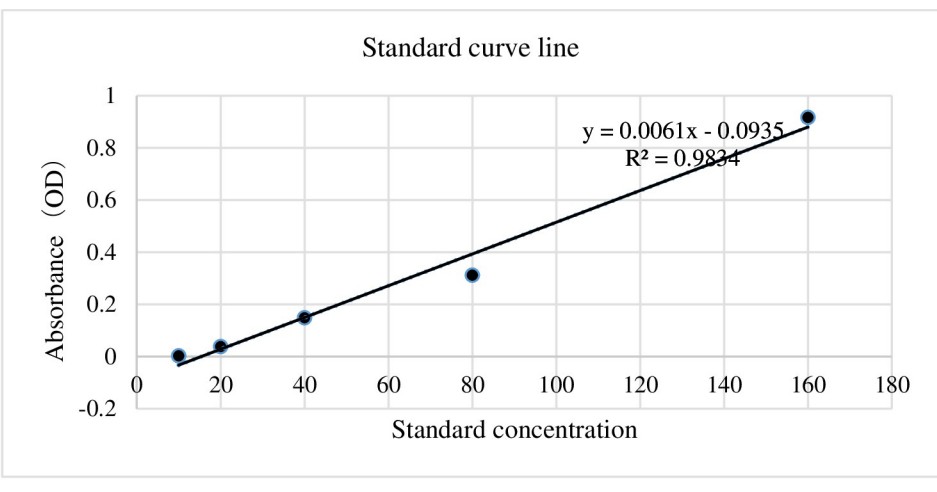

A

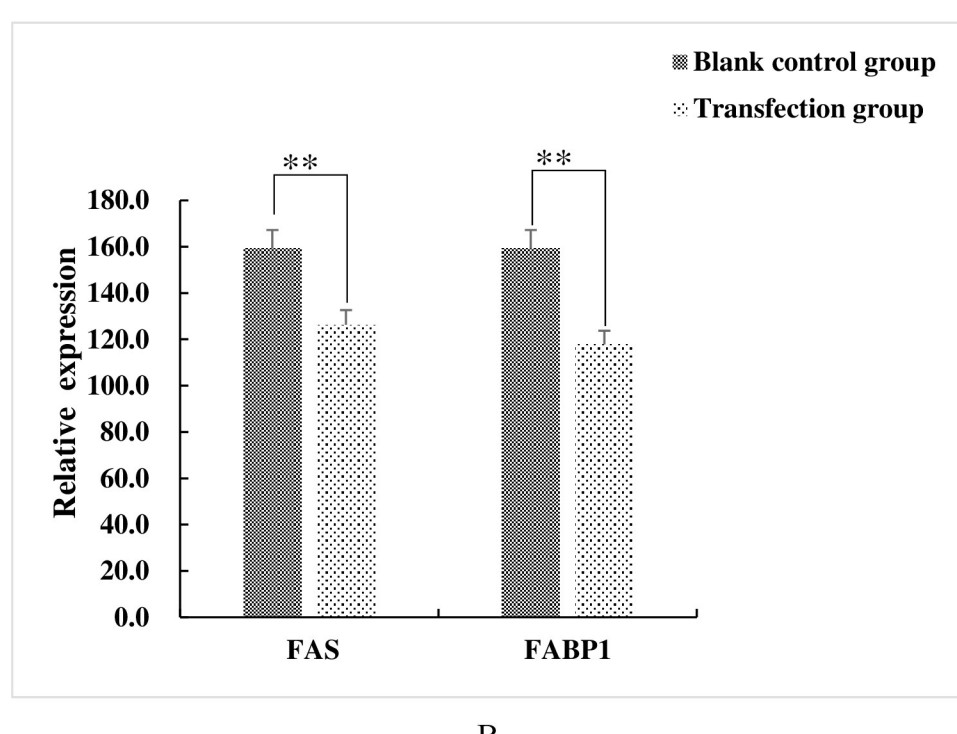

B

**Fig 5.** Lysyl oxidase activity is decreased in cultured adipocytes over-expression either FAS of FABP1 (A) Standard curve for lysyl oxidase activity. (B) LOX activity measured in cells transfected with FAS and FABP1 as indicated. **, p < 0.01.

mirrored those of the *mRNA* levels and LOX enzyme activity decreased to levels that were significantly (P < 0.01) lower for the transfectants compared with controls (Fig 5B and 5C).

## 4 Discussion

The FABP gene family encodes fatty acid-binding proteins that are involved in the transport of intracellular fatty acids as well as the synthesis and degradation of fat in the body. Therefore, this family plays an important role in the fat metabolism in the body [18]. FABP1 plays an

important regulatory role in fat deposition and participates in the formation of triglycerides [19]. Elevated expression of FABP1 in porcine preadipocytes significantly increases fat accumulation [20, 21]. In goats, the FABP1 gene in subcutaneous fat is also expressed at significantly higher levels than in the control *longissimus dorsi* muscle [16]. FAS is a key enzyme in fat synthesis in animals and promotes fatty acid formation. The FAS gene is expressed in mammalian fat, kidney and lungs and other tissues and cells [22]. FAS is also highly expressed in the subcutaneous adipose tissue of Baixi and first generation Subai hybrid pigs [23] in addition to Bamei and large white breeds [24, 25]. In this study, we found that FABP1 and FAS gene expression were significantly higher in subcutaneous fat than for the control muscle tissue. It is speculated that FABP1 and FAS may be used as candidate genes for fat deposition studies especially since higher expression levels correlated with greater fat deposition. Our in-silico bioinformatic analysis of FABP1 and FAS proteins indicated slight differences but the overall structures were similar to their counterparts in other pig breeds. At the same time, alterations in the primary sequences that would affect enzymatic functions were not detected [26].The expression level of FABP1 and FAS in muscle and adipose tissue is different among breeds and within breeds [24, 25]. The expression level of subcutaneous adipose tissue in Kele pig and DLY crossbred pig is significantly higher than *longissimus dorsi* muscle. Moreover, the expression level of subcutaneous adipose FABP1 in the Kele pig is higher than DLY crossbred pig, which may be due to the fact that the Kele pig is a local pig that present a high fat deposition [12]. In this study, the expression of FABP1 and FAS genes in the subcutaneous adipose tissue of Zongdihua pig is significantly higher compared to *longissimus dorsi* muscle, which is similar to other pig breeds. These data may indicate that FABP1 and FAS could be used as genes to monitor fat deposition and higher expression levels are associated with stronger fat deposition ability.

The study shows that the expression of the COL3A1 gene in pig subcutaneous fat is significantly higher compared to *longissimus dorsi* muscle. The results were consistent with higher levels of collagen in fat compared to muscle [27]. The expression of LOX is lower in fat compared to muscle, and the content of pyridine cross-linking (HP and LP) is the lowest in subcutaneous fat [27]. We speculate that this might be related to fat deposition. By culturing primary adipocyte from subcutaneous fat we were able to successfully over-express FAS and FABP1 in Zongdihua pig preadipocytes and this resulted in increased COL3A1 and decreased LOX expression levels as well as lysyl oxidase activities These results indicate that fat deposition may disrupt the structure of connective tissues resulting in an increase in tenderness with age [28]. Therefore, fat deposition may promote COL3A1 gene expression thereby promoting collagen production and inhibiting LOX gene expression. Over-expression of FAS can promote cell proliferation, inhibit apoptosis and elevate triglyceride synthesis in adipocytes. FAS is a key regulator of fat deposition and its over-expression can promote cell proliferation, inhibit cell apoptosis and also affect the synthesis of triglycerides in adipocytes. Therefore, these FAS gene products collectively affected the deposition and composition of fat [29].

Collagen is an ECM protein that not only supports adipocytes, but also regulates the development of adipose tissue via cell-cell signaling [30]. Disruption of collagen III synthesis may lead to impaired triglyceride accumulation in adipocytes and disrupt adipose tissue remodeling [31]. A previous study has demonstrated a role for adipose tissue in regulating insulin sensitivity that can also enhance the beneficial metabolic effects of branched-chain fatty acid esters of hydroxy fatty acids, and therefore enhance anti-inflammatory effects [32]. Lysyl oxidase (LOX) and 4 lysyl oxidase-like proteins (LOXL 1–4) are copper amine oxidases with a highly conserved catalytic domains that use lysine tyrosyl quinone cofactors and possess conserved copper binding sites. This group also catalyzes the first step in covalent cross-linking of ECM proteins such as collagen and elastin that are responsible for mechanical strength of the

ECM [33–35]. Our results indicated that over-expression of FAS and FABP1 genes significantly increased the expression of COL3A1 gene while inhibited LOX gene. This led to the conclusion that cellular FAS and FABP1 levels can regulate collagen accumulation. However, fat deposition can also promote collagen synthesis but this is negatively correlated with the expression of lysyl oxidase activity and *mRNA* accumulation. Overexpression of FAS and FABP1 genes in our study increased *COL3A1* expression while decreasing LOX expression indicating that fat deposition is accompanied by increased collagen synthesis and decreased LOX activity resulting in a decrease in mature cross-linking content. Therefore, the elevated expression of these genes in Zongdihua pigs generated high levels of fat and collagen and low levels of collagen cross-linking resulting in thick and waxy skin and increased meat tenderness [36].

## 5 Conclusions

The current study demonstrated that the *FAS* and *FABP1* gene expression was higher in subcutaneous fat of Zongzi pigs compared with muscle tissues. Over-expression of FAS and *FABP1* significantly increased expression of *COL3A1* and significantly inhibited LOX expression and its corresponding lysyl oxidase activity. Our findings suggest that the *FAS* and *FABP1* genes may be fat-related candidate gene targets. The next step will be to further explore the relationships between collagen and its structural properties with meat quality traits in Zongdihua pigs and to explore the function of the collagen and *LOX* gene families at the molecular level. An in-depth genetic exploration of the internal regulatory mechanism of collagen synthesis and its cross-linking properties may reveal specific regulatory and signaling pathways that can be used to improve the genetics of this animal used for food production.

## Supporting information

**S1 Fig.** PCR amplification of (A) FAS and (B) FABP1.
(TIF)

**S2 Fig. Morphological changes of pre-adipocyte primary cultures.** Photomicrographs of cultures taken after (A) 4 h (B) 24 h (C) 48 h and (D) 72 h. 100× magnification.
(TIF)

**S3 Fig. Induced differentiation of intramuscular preadipocytes.** Photomicrographs of differentiating cells at (A) 24 h (B) 4 d and (C) 6 d. (D) Oil red O staining of a cell monolayer after 5d in differentiation medium. 100 × magnification.
(TIF)

**S4 Fig.** (A) pEGFP-C1-*FAS* transfected cells (B) vector only (C) blank control. (D) pEGFP-C1-*FAS* transfected cells (E) vector only and (F) blank control.
(TIF)

**S5 Fig.** The amplification curve and melting curve of (A)*COL3A1*, (B)*LOX*, (C)*FAS* gene.
(TIF)

**S6 Fig.** The amplification curve and melting curve of (D)*FABP1*, (E)*GAPDH* gene.
(TIF)

## Author Contributions

**Conceptualization:** Rong Yang.

**Data curation:** Rong Yang, Di Zhou, Zhihong Yan.

**Formal analysis:** Rong Yang, Di Zhou, Zhihong Yan, Xin Wang.

**Funding acquisition:** Rong Yang.

**Investigation:** Rong Yang, Yan Wang.

**Methodology:** Rong Yang, Zhonghai Zhao.

**Project administration:** Rong Yang, Zhihong Yan.

**Resources:** Rong Yang, Lingling Xie, Xin Wang.

**Software:** Rong Yang, Lingling Xie.

**Supervision:** Rong Yang, Jun Li, Liqun Ren.

**Validation:** Rong Yang, Jun Li, Liqun Ren.

**Visualization:** Rong Yang.

**Writing – original draft:** Rong Yang.

**Writing – review & editing:** Di Zhou.

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
