## [Decision Letter · Decision Letter 0]

15 Jul 2022

PONE-D-22-15349Fatty acid binding protein 1 and fatty acid synthetase overexpression have differential effects on collagen III and cross-linking in Zongdihua pig tissuesPLOS ONE

Dear Dr. Zhou,

Thank you for submitting your manuscript to PLOS ONE. After careful consideration, we feel that it has merit but does not fully meet PLOS ONE’s publication criteria as it currently stands. Therefore, we invite you to submit a revised version of the manuscript that addresses the points raised during the review process.

We look forward to receiving your revised manuscript.

Kind regards,

Marcio Duarte, PhD

Academic Editor

PLOS ONE

Journal Requirements:

4. PLOS requires an ORCID iD for the corresponding author in Editorial Manager on papers submitted after December 6th, 2016. Please ensure that you have an ORCID iD and that it is validated in Editorial Manager. To do this, go to ‘Update my Information’ (in the upper left-hand corner of the main menu), and click on the Fetch/Validate link next to the ORCID field. This will take you to the ORCID site and allow you to create a new iD or authenticate a pre-existing iD in Editorial Manager. Please see the following video for instructions on linking an ORCID iD to your Editorial Manager account: https://www.youtube.com/watch?v=_xcclfuvtxQ.

Additional Editor Comments:

Please double check the statistical analysis description. In your revision, make sure to describe in detail what analysis were performed. Also, please include in the revised manuscript the equation of your experimental model for better clarification of what were the effects considered on statistical analysis. The lack of the detail description in statistical analysis may result in the rejection of the manuscript.

Reviewers' comments:

Reviewer's Responses to Questions

**Comments to the Author**

1. Is the manuscript technically sound, and do the data support the conclusions?

Reviewer #1: Partly

Reviewer #2: Yes

Reviewer #3: Partly

2. Has the statistical analysis been performed appropriately and rigorously? 

Reviewer #1: I Don't Know

Reviewer #2: I Don't Know

Reviewer #3: No

3. Have the authors made all data underlying the findings in their manuscript fully available?

Reviewer #1: Yes

Reviewer #2: Yes

Reviewer #3: Yes

4. Is the manuscript presented in an intelligible fashion and written in standard English?

Reviewer #1: Yes

Reviewer #2: Yes

Reviewer #3: No

5. Review Comments to the Author

Reviewer #1: Title: Fatty acid binding protein 1 and fatty acid synthetase overexpression have differential

effects on collagen III and cross-linking in Zongdihua pig tissues

The current manuscript reports the initial basis on how the overexpression of FABP1 and FAS influence fat accumulation in adipocytes and in the expression of COL3A1 and LOX. The results may serve to other more complexes studies, and these future perspectives should have been reported. Moreover, an additional attention should be given to the manuscript before submission to avoid typos and the inclusion of incorrect references. In summary, my concerns are a) It was not reported the number of animals used for tissue extraction; b) the lack of protein abundance analysis to evaluate the functionality of the evaluated genes (mainly FAS, FABP1, COL3A1); c) the lack of references supporting the presence of FABP isoform 1 in muscle and adipocytes. Please consider the comments below.

Introduction

-The first citation starts in 3. Please review the references and citations.

-Why FABP1 was chosen to be evaluated instead of other known muscle/adipocyte FABPs isoforms? Previous studies have reported FABP1 as a predominant isoform in liver. If there is any evidence reporting FABP1 in muscle/adipocytes I suggest adding such information in introduction and/or discussion section.

L42-43: change “that is synthesized in large amounts” to “extensively synthesized”

L44- 47: change “reservoir of” to “storage of”

L52-55: This paragraph needs to be improved and the aim of the study need to be consisting to what was evaluated. The current study does not provide sufficient molecular and genetic basis for such affirmation.

Material and methods

L58: Typo “Statement”

L64: Please specify how many pigs were used in each age

L65: Correct the name of pigs’ breed

L68: “…, frozen in liquid nitrogen and stored at -80 °C to further analyses.”

L69, L81: Please unify the information from those sections to its respective analysis

L88: Which shaded areas?

L102: Please cite the reference of this method

L105: Subcutaneous fat cDNA from which pigs? 3d and 9m pigs?

L109: Please specify here the name of the commercial kit

L118: How many 3d old pigs were used? The previous analysis (genes expression) was done with 3d and 9m old pigs? Please provide an explanation about the use of 3 d old pigs for adipocytes culture instead of 9m old pigs.

L145: Please specify here the name of the commercial kit

Please avoid the use of “(see below)” and “(see above)” throughout the text.

L148: the section 1.5 can be placed together with this section

L153: Please add the statistical model in this section.

L154: Please cite the method used to normalize mRNA

L157: Why DNA sequencing was performed? Please specify and add the methods and steps of this analysis.

Results

L159: Please remove “analysis” from the section’s title

L169-172: Please indicate what was the purpose of these analyses.

Figure 4: Please verify figure’s letters

L172-174: Please explain how this analysis was performed?

L179-L181: Please correct the figure citation

Figure 6: intramuscular or subcutaneous preadipocytes?

L195: “…were constructed and fused to …”

L199 and figure 7: what would be the blank control? please specify

Fig 8: Please correct figure legend “..transfected with pEGFP-C1-FAS and pEGFP-C1-FAS as indicated..”

L204 and Figure 8: Please specify what would be the control group.

L208-209: Did you mean double-antibody sandwich ELISA method?

Discussion

L2017-218: Is there any other reference to cite this statement? I could not find reference number 13. Also, the “adipocyte-type fatty acid binding protein” in the title referees to A-FABP or FABP4 and not FABP1.

L219-220: Reference number 14 did not evaluate FABP1 expression.

L220-221: Is there any other reference to cite this statement? I could not find reference number 15. Please carefully revise the discussion section and if possible, change the references that are not written in English, so the reader can easily find in the future.

L226-228: This is a strong affirmative that need to be rewritten. Since the current study did not evaluate the protein levels of FABP1 and FAS or used methods to inactive the function of these proteins, there is no assurance that FABP1 and FAS can be considered the only candidate genes to infer in fat accumulation.

L228-230: The results of this analysis need to be shown and better explained, as well as the methodology.

L232: Please cite the previous studies.

L232: “....that COL3A1 and LOX…”

L233: “…than longissimus dorsi muscle ...”

L232-233: That is not what was reported in Figure 1, where the expression of all the genes were higher in subcutaneous fat compared to muscle.

L234: Please add a reference to the statement.

L236: What did you mean with “may damage”? Please provide better explanation

L243: Typo “adipocytes”

L244: Which gene products? What did you mean with “decomposition of fat”. Please be more objective

L253: Add citation

L262: Is there any refence reporting the meat quality parameter of Zongdihua pigs comparing with other breeds? If so, it could be included to support this hypothesis.

Conclusions

As previously mentioned, this affirmation should be excluded or reformulated. Moreover, a perspective of future studies could be included.

Figures

I suggest reporting Figures 2, 5, 6 , and 7 as supplementary.

Reviewer #2: TITLE

The title reflects the work's main result which means that instead of “Zongdihua pig tissues” you must use Zongdihua pig isolated adipocytes once the overexpression was done in isolated cells, not in tissue.

ABSTRACT

Line 19. Please, Replace Zongdihua pig by Zongdihua pig isolated adipocyte

Line 19. Replace identify processes by identify biochemical processes

Line 20. Replace, using molecular genetics by molecular tools

INTRODUCTION

Line 40. In the first paragraph, please, develop in a deeper way the signaling pathway crosslink between the genes FABP1, FAS, and COL3A1

Line 53. Please, in the sentence “In the present study… you should increment the paragraph with a clear objective of the study. Which question will be answered?

MATERIAL AND METHODS

Line 87. Please, add the Accession number (e.g NM_) of the sequences you have used to design oligonucleotides.

Line 138. If you have used 4ug of plasmid to transfect, please add this information just before the sentence “recombinant plasmids pEGFP-C1-FABP1 and pEGFP-C1-FAS”. Otherwise, what is the 4ug of DNA? Add empty plasmids controls and blanks to sentence. Clarify this sentence until line 140.

Line 153. In Data processing and analysis, The 2-△△Ct method is a gene expression method and I do not identify the statistical methods used not only in gene expression analysis, but also in the other analysis. Please, add the proper information and experimental design in this subtopic.

RESULTS AND ANALYSIS

Line 163. Please, delete the term “extremely”

Line184. Delete the term “and differentiate” or changed it to “morphology differentiate” following the idea of the next sentence.

Line 191. Don’t you have the quantification the lipids content staining with Oil Red O as you mentioned in MM? Put the data in figure 6 and refer to it here.

Line 195. Please, the sentence “Cloned versions of FAS and FABP1 were constructed that were fused to the eGFP protein at their carboxyl termini and over-expressed via the strong CMV promoter” is important to MM section. Relocate it. Otherwise, just here the reader will understand what was done.

DISCUSSION

It was well developed

CONCLUSION

It was well developed

FIGURES

Figures 1 and 8. Please, in the Y axis add “/GAPDH” just after relative expression.

Figures 5 and 6. Please, add magnification bar in the bottom corner of the image. Images seems in different magnification.

Figure 6. Please, add E image of graphical representation of the Oil red O staining quantification as you mentioned on MM.

Figure 7. Here we have some problems. This figure per se is not a result, but only a control of the efficiency of you transfection. I strongly recommend you to put together this figure to figure 8. In addition, I think we have a problem with the image once only pEGFP-C1-FAS transfected cell have been shown (doubled data). The magnification of the images seems different, please, correct that.

Reviewer #3: Dear authors,

The manuscript entitled “Fatty acid binding protein 1 and fatty acid synthetase overexpression have differential effects on collagen III and cross-linking in Zongdihua pig tissues” aims to determine whether FABP1 and FAS regulate expression of collagen and its crosslinking via lysyl oxidase in Zongdihua pigs. The manuscript points out that enhancing the expression of FAS and FABP1 increases collagen accumulation and this preliminarily suggests that FAS and FABP1 can serve as fat-related candidate genes providing a theoretical basis for the study of fat deposition in Zongdihua pigs. The manuscript brings interesting results, however, to be accepted for publication, it needs major changes and corrections. Below are suggestions to make the manuscript suitable for publication and more informative to the reader.

General suggestions

- Write all gene names in italics. Example: FABP1

- Please review the text for language.

- Some words are used in different ways in the manuscript such as cross-linking/crosslinking, Over-expression/Overexpression…Correct this, please.

- The introduction needs to be rewritten, addressing the issues related to the manuscript more deeply. In addition, give a clear hypothesis and objective in the introduction section.

- Please, change “longissimus dorsi” to “Longissimus dorsi” throughout the manuscript.

- Discussion needs to be rewritten, addressing more deeply the observed results.

Abstract

- Line 26 - Remove the with in the sentence "FAS encodes 333 amino acid hydrophobic protein containing with 26 phosphorylation sites and 0 transmembrane regions".

- Line 27 - Change the following sentence "The basal levels of FABP1 and FAS in pig tissues expression were 3 -3.5-fold higher in subcutaneous fat compared with muscle (P < 0.01)" to "The basal levels of FABP1 and FAS expression in pig tissues were 3 -3.5-fold higher in subcutaneous fat compared with muscle (P < 0.01)".

- Line 33- I believe you forgot to add “FAS” in this sentence "Therefore, enhancing FABP1 expression increases collagen accumulation...". Am I correct? Because the overexpression of FAS and FABP1 resulted in a significantly increased expression of the COL3A1 gene.

- Keywords must be arranged in alphabetical order. It is not recommended to add words that are already in the title of the manuscript such as Zongdihua pig.

Introduction

- Add the title “Introduction” before the text in the line 40.

- You could improve your introduction by talking more about the breed (Zongdihua pig), mainly about the meat traits. It could also present some preliminary studies that address the evaluated genes deeply. All these issues need to be better integrated. In addition, the objective of the work should be better presented in this introduction.

Materials and methods

- Line 58 - Correct to Ethics Statements.

- Why is the topic "Ethics Statements" highlighted in the text?

- Please, I ask you to correct the text in relation to the language used. Some sentences are poorly worded.

- You don't write about the design and accession number of the following genes: LOX and COL3A1 in the text.

- Did you work with the fold change (comparative CT method)? Why are the graphs with the Y-axis legend as a relative expression? I suggest reading the article "Analyzing real-time PCR data by the comparative CT method - Thomas D Schmittgen1 & Kenneth J Livak 2008" to correct this question.

- Subcutaneous fat and Longissimus dorsi samples were collected from how many animals? What is the N of this gene expression analysis? You need to better describe the experimental design of the study.

Results

- Why in the legend of figure 1 you used fat deposition if there is only information about the fold change between subcutaneous fat and Longissimus dorsi?

- Line 163:164 – “The expression levels for all 4 genes were extremely significantly (P < 0.01) elevated in subcutaneous fat tissues compared with the muscle (control) tissues (Figure 1).” Why did you compare subcutaneous fat to muscle? Your hypothesis should be very clear in the manuscript in relation to the comparisons made. Besides that, could you explain a little more about what you considered as a control in this study?

- Is the Y-axis legend correct in Figure 9?

Discussion

- Line 225:226 – Change the sentence “In this study, we found that FABP1 and FAS gene expression were significantly higher than for the control muscle tissue” to “In this study, we found that FABP1 and FAS gene expression were significantly higher in subcutaneous fat than for the control muscle tissue”

- Line 232 – Alter LOC gene to LOX.

- Your discussion of the results is superficial, you need to go deeper into the effect of the FAS and FABP1 genes on the expression of COL3A1 and LOX and the production of collagen. In addition to the effects of these genes on the deposition of fat and on meat traits.

- Line 231:233- “Our results agreed with previous studies indicating that COL3A1 and LOC gene expression in subcutaneous fat are significantly higher and lower than in longissimus dorsi muscle tissues, respectively.” - Figure 1 shows greater expression of both the COL3A1 and LOX genes in subcutaneous fat in relation to Longissimus dorsi.

- You have not discussed deeply the effects of the expression of these genes on meat quality in the breed studied.

6. PLOS authors have the option to publish the peer review history of their article (what does this mean?). If published, this will include your full peer review and any attached files.

Reviewer #1: No

Reviewer #2: **Yes: **Da Silva, Walmir

Reviewer #3: No

---

## [Author Response · Author response to Decision Letter 0]

14 Sep 2022

Responses to Reviewers’ comments

Comments to the Author

1. Is the manuscript technically sound, and do the data support the conclusions?

Reviewer #1: Partly

Reviewer #2: Yes

Reviewer #3: Partly

Responses from the authors: We acknowledge the comments. Thanks!

2. Has the statistical analysis been performed appropriately and rigorously? 

Reviewer #1: I Don't Know

Reviewer #2: I Don't Know

Reviewer #3: No

Responses from the authors: Acknowledged.

3. Have the authors made all data underlying the findings in their manuscript fully available?

Reviewer #1: Yes

Reviewer #2: Yes

Reviewer #3: Yes

Responses from the authors: Thanks!

4. Is the manuscript presented in an intelligible fashion and written in standard English?

Reviewer #1: Yes

Reviewer #2: Yes

Reviewer #3: No

Responses from the authors: We have our revised manuscript been proofread by a native English-speaking editor from a professional editing service.

Reviewers’ Comments to the Author 

Reviewer #1: 

Title: Fatty acid binding protein 1 and fatty acid synthetase overexpression have differential

effects on collagen III and cross-linking in Zongdihua pig tissues

The current manuscript reports the initial basis on how the overexpression of FABP1 and FAS influence fat accumulation in adipocytes and in the expression of COL3A1 and LOX. The results may serve to other more complexes studies, and these future perspectives should have been reported. Moreover, an additional attention should be given to the manuscript before submission to avoid typos and the inclusion of incorrect references. In summary, my concerns are a) It was not reported the number of animals used for tissue extraction; b) the lack of protein abundance analysis to evaluate the functionality of the evaluated genes (mainly FAS, FABP1, COL3A1); c) the lack of references supporting the presence of FABP isoform 1 in muscle and adipocytes. Please consider the comments below.

Responses from the authors: Thanks for your excellent summary comments. Your points are well taken. Per your suggestion, in the revised manuscript, we added some future perspectives in the Discussions/Conclusions sections. The revised manuscript has been proofread by a native English speaker to avoid typos and grammar misuses. 

Regarding your concerns, a) we have added the number of animals used for tissue extraction explements; b) regarding protein abundance analysis, we highly agree with your suggestion. It was considered at the time; however, it was in January of 2020, the COVID-19 outbreak occurred in China. Still, the COVID control policy makes it difficult to conduct experiments. We will keep this in mind in our follow-up studies in the future; and c) thanks for the comment. We have added the following in the Introduction section with new references to support the presence of FABP isoform 1 in muscle and adipocytes. It reads in the revised manuscript (the actual numbering of the references will be adjusted in the manuscript): 

“FABP1 and FAPB2 genes play an important role in the regulation of fat metabolism, and FABP1 gene is expressed in various tissues of pigs [1]. Previous study showed that duck L-FABP or FAPB1 gene polymorphism was significantly correlated with breast muscle fat content, and expressed in other tissues [2]. Thus, it was speculated that the L-FABP gene may be the main gene that affects pig intramuscular fat content or a marker gene closely linked to the main gene [3].”

[1] Xiong Xun, Li Yong, Ruan Yong, et al. Molecular characteristics and tissue expression analysis of FABP1 and FABP2 genes in pigs [J]. Southern Agricultural Journal, 2022, 53(4):11.

[2] He Jun. Correlation between L-FABP, A-FABP gene polymorphisms and intramuscular fat content in ducks and their effects on the expression of genes related to fat metabolism [D]. Zhejiang University, 2013.

[3] Cui Lu, Chen Xing, Zhou Ying, et al. Research progress of FABPs family genes [J]. Animal Husbandry and Veterinary Medicine, 2016(7):4.

Introduction

-The first citation starts in 3. Please review the references and citations. 

Responses from the authors: Thanks for this comment. The reference list has been checked and fixed. 

-Why FABP1 was chosen to be evaluated instead of other known muscle/adipocyte FABPs isoforms? Previous studies have reported FABP1 as a predominant isoform in liver. If there is any evidence reporting FABP1 in muscle/adipocytes I suggest adding such information in introduction and/or discussion section. 

Responses from the authors: Thanks for the comment. To our knowledge, FABP1 and FAPB2 genes are two FABPs genes discovered earlier, which play an important role in the regulation of fat metabolism [1], and FABP1 gene is expressed in various tissues of pigs. He Jun et al [2] showed that duck L-FABP (FAPB1) gene polymorphism was significantly correlated with breast muscle fat content, expressed in other tissues, and played an important regulatory role in fat metabolism; thus, it could affect a variety of expression of fat metabolism-related genes. Cui Lu et al. [3] speculated that the L-FABP gene may be the main gene that affects pig intramuscular fat content or a marker gene closely linked to the main gene. Therefore, in this study, we selected FABP1.

[1] Xiong Xun, Li Yong, Ruan Yong, et al. Molecular characteristics and tissue expression analysis of pig FABP1 and FABP2 genes. Southern Agricultural Journal, 2022, 53(4):11.

[2] He Jun. Correlation between L-FABP, A-FABP gene polymorphisms and intramuscular fat content in ducks and their effects on the expression of genes related to fat metabolism. Zhejiang University, 2013.

[3] Cui Lu, Chen Xing, Zhou Ying, et al. Research progress of FABPs family genes. Animal Husbandry and Veterinary Medicine, 2016(7):4.

L42-43: change “that is synthesized in large amounts” to “extensively synthesized”

Responses from the authors: Thanks for this suggestion. Per your comment, it has been changed according.

L44- 47: change “reservoir of” to “storage of”

Responses from the authors: Per suggestion, it has been changed according.

L52-55: This paragraph needs to be improved and the aim of the study need to be consisting to what was evaluated. The current study does not provide sufficient molecular and genetic basis for such affirmation.

Responses from the authors: Thanks for your comment. In this study, the expression levels of FABP1 and FAS genes in the longissimus dorsi and subcutaneous adipose tissue of Zongdihua pigs were detected by qPCR technology, and the eukaryotic expression vectors of pEGFP-C1-FAS and pEGFP-C1-FABP1 were constructed to detect the difference between FABP1 and FAS. The role of collagen and its pyridine cross-linking, to explore the mechanism of fat deposition on collagen and cross-linking, and to provide a molecular theoretical basis for improving the resources of Zongdihua pig breeds.

Per your suggestion and those of other reviewers, we have added more background information in the Introduction section in the revised manuscript. 

Material and methods

L58: Typo “Statement”

Responses from the authors: Corrected. Thanks.

L64: Please specify how many pigs were used in each age. 

Responses from the authors: There were 6 for each age group. This information has been edited in the revised manuscript.

L65: Correct the name of pigs’ breed

Responses from the authors: Thanks for the comment. We have added the breed (Zongdihua pig) in the revised manuscript.

L68: “…, frozen in liquid nitrogen and stored at -80 °C to further analyses.”

Responses from the authors: Thanks for the comment, and we have changed it in the text.

L69, L81: Please unify the information from those sections to its respective analysis

Responses from the authors: We have revised it accordingly. Thanks.

L88: Which shaded areas? 

Responses from the authors: They refer to shaded nucleotides of primer sequences of FABP1-CDS and FAS-CDS in Table 1. The revised manuscript reflects the changes accordingly.

L102: Please cite the reference of this method

Responses from the authors: Thanks, and we added a reference for this.

[4] Yang Yang, Xu Houqiang, Chen Wei, et al. Construction of Congjiang Xiang pig FGF10 gene interference vector and analysis of related gene expression. Journal of Agricultural Biotechnology, 2018, 26(12):9.

L105: Subcutaneous fat cDNA from which pigs? 3d and 9m pigs?

Responses from the authors: They were from the 3-day old pigs. It has been added to the text. 

L109: Please specify here the name of the commercial kit

Responses from the authors: Thanks for this comment. We have added the information in the text. 

L118: How many 3d old pigs were used? The previous analysis (genes expression) was done with 3d and 9m old pigs? Please provide an explanation about the use of 3 d old pigs for adipocytes culture instead of 9m old pigs

Responses from the authors: Thanks for the comments. There were six 3-day-old pigs used in the study. In the gene expression study, we used six of 9-month-old pigs. Since cells from younger pigs (such as 3d pigs) have better proliferation and differentiation ability, we used 9-month-old pigs for adipocytes culture, as they are more matured and have less cell proliferation and differentiation ability, yielding fewer variable cultures.

L145: Please specify here the name of the commercial kit Please avoid the use of “(see below)” and “(see above)” throughout the text.

Responses from the authors: Thanks for the comments. Our points are well taken. We have fixed them in the revised manuscript. 

L148: the section 1.5 can be placed together with this section

Responses from the authors: Thanks for this comment. Now, section 1.9 has been merged to section 1.5.

L153: Please add the statistical model in this section.

Responses from the authors: Thanks for the comment. The 2-△△Ct statistic method was used to analyze the experimental data, and SPSS 20.0 software was used to analyze the significance level. This has been clarified in the text. 

L154: Please cite the method used to normalize mRNA.

Responses from the authors: Thanks for the comment. For the method used for normalizing mRNA, we have added the following reference in the text.

Yang Yang, Xu Houqiang, Chen Wei, et al. Construction of Congjiang Xiang pig FGF10 gene interference vector and analysis of related gene expression. Journal of Agricultural Biotechnology, 2018, 26(12):9.

L157: Why DNA sequencing was performed? Please specify and add the methods and steps of this analysis. 

Responses from the authors: Thanks for the comments. DNA sequencing was mainly to verify whether the eukaryotic vector was successfully constructed. Per your suggestion, we have added the information to the text. 

Results

L159: Please remove “analysis” from the section’s title

Responses from the authors: Thanks for the comments. Done, it is removed.

L169-172: Please indicate what was the purpose of these analyses

Responses from the authors: The purpose of comparing the cloned fragment with the original sequence was to verify that the cloned fragment is the desired fragment or the correct fragment. Additionally, the bioinformatics analysis (original lines 170-172) was mainly to predict the physicochemical properties of FAS proteins and to understand the locality of Guizhou porcine FAS protein structure

Figure 4: Please verify figure’s letters

Responses from the authors: Thanks for the reminder. They are checked in the revised manuscript.

L172-174: Please explain how this analysis was performed? 

Responses from the authors: Analysis of the secondary structure and physicochemical properties of FAS and FABP1 proteins were performed using the Protean program in DNAstar software. The text in the revised manuscript is changed to reflect this.

L179-L181: Please correct the figure citation.

Responses from the authors: Thanks for this comment. They are checked.

Figure 6: intramuscular or subcutaneous preadipocytes?

Responses from the authors: It referred to subcutaneous preadipocytes. 

L195: “…were constructed and fused to …”

Responses from the authors: Thanks for the comment. It is fixed in the text.

L199 and figure 7: what would be the blank control? please specify. 

Responses from the authors: Good comment. Blank control refers to the group of cells not transfected with pEGFP-C1-FAS and pEGFP-C1-FABP1 eukaryotic vectors. This has been reflected in the revised manuscript.

Fig 8: Please correct figure legend “...transfected with pEGFP-C1-FAS and pEGFP-C1-FAS as indicated...”

Responses from the authors: Thanks for the comment. It is modified in the figure legend.

L204 and Figure 8: Please specify what would be the control group. 

Responses from the authors: Thanks for the comment. In the figure, the data from control group are shown in the bar on the left of each gene product, as they are marked in red below.

L208-209: Did you mean double-antibody sandwich ELISA method? 

Responses from the authors: Thanks for the comment. Yes, it is ELISA method.

Discussion

L217-218: Is there any other reference to cite this statement? I could not find reference number. Also, the “adipocyte-type fatty acid binding protein” in the title referes to A-FABP or FABP4 and not FABP1.

Responses from the authors: Thanks for the good comment. We have added the following reference. 

Ai Jinxin, Long Anju, Luo Weixing, et al. Study on the expression levels of FABP1 and FABP3 genes in different tissues of Qianbei hemp sheep. China Animal Husbandry, 2020, 56(11):68-72+78

L219-220: Reference number 14 did not evaluate FABP1 expression

Responses from the authors: Thanks for pointing this out. We have replaced it with the following two references: 

1）Chen Guilian. Genetic polymorphism of pig L-FABP gene and its correlation with intramuscular fat content. Hebei Agricultural University, 2008. 

2）Gao Yan, Zhang Yonghong, Xu Yanli , et al. Exon 2 polymorphism of L-FABP gene in Songliao black pigs and its correlation with meat quality traits. Chinese Journal of Veterinary Medicine, 2010, 30(8):4.

L220-221: Is there any other reference to cite this statement? I could not find reference number 15。 Please carefully revise the discussion section and if possible, change the references that are not written in English, so the reader can easily find in the future

Responses from the authors: Thanks for the comment. We have made sure that original reference 15 is listed in the revised manuscript, which is:

Ai Jinxin, Long Anju, Luo Weixing, et al. Study on gene expression levels of FABP1 and FABP3 in different tissues of Qianbei hemp sheep [J]. Chinese Journal of Animal Husbandry, 2020, 56(11):68- 72+78. It is 18 in the revised manuscript.

L226-228: This is a strong affirmative that need to be rewritten. Since the current study did not evaluate the protein levels of FABP1 and FAS or used methods to inactive the function of these proteins, there is no assurance that FABP1 and FAS can be considered the only candidate genes to infer in fat accumulation.

Responses from the authors: Thanks for the comment. We agree with your point and made a modification to the sentence in the revised manuscript.

L228-230: The results of this analysis need to be shown and better explained, as well as the methodology.

Responses from the authors: Thanks for the comment. Per your suggestion, we added the following in the Discussion section, which reads:

In this study, the expression levels of FABP1 and FAS genes in the muscle and adipose tissue of Zongdihua pigs were detected by qRT-PCR method. muscle (P<0.01), which is the same as the previous research results, and the expression specificity tends to be consistent [18-19]. It indicated that FABP1 and FAS could be used as candidate genes for fat deposition. The higher the expression level, the stronger the fat deposition ability. Subsequently, DNAstar software was used to analyze the FABP1 and FAS proteins of Zongdihua pigs. Therefore, we used subcutaneous adipose tissue to culture primary adipocytes for more targeted and comprehensive experiments.

L232: Please cite the previous studies.

Responses from the authors: Thanks for pointing this out. References are added (25).

L232: “....that COL3A1 and LOX…”

Responses from the authors: Thanks for the comment. Together with consideration of another review, we have deleted LOX. 

L233: “…than longissimus dorsi muscle ...”

Responses from the authors: Similar to your comment immediately above, we have fixed this part in the revised manuscript. 

L232-233: That is not what was reported in Figure 1, where the expression of all the genes were higher in subcutaneous fat compared to muscle.

Responses from the authors: Thanks for the comment. It is a very good catch. The sentence now is revised as you suggested.

L234: Please add a reference to the statement. 

Responses from the authors: Thanks for the comment. We have added the following reference that its finding showed that the expression level of COL3A1 gene in subcutaneous fat was significantly higher than that in longissimus dorsi muscle tissue.

Yang Rong, Zhou Di, Yan Zhihong, et al. Research on the characteristics of local pig collagen in Guizhou and the tissue expression of related genes [J]. Chinese Journal of Animal Husbandry, 2021, 57(8):5: 

L236: What did you mean with “may damage”? Please provide better explanation

Responses from the authors: Thanks for the comment. We meant to say that fat deposits may disrupt the structure of connective tissue. It has been modified in the revised manuscript.

L243: Typo “adipocytes”

Responses from the authors: Thanks for pointing out the typo. It has been corrected.

L244: Which gene products?

Responses from the authors: Thanks for the comment. It refers to FAS gene products, as FAS gene expression can affect fat deposition and breakdown. We have clarified in the text.

What did you mean with “decomposition of fat”. Please be more objective.

Responses from the authors: Thanks for the comment. It is because that FAS gene is believed a key gene in the process of fat deposition, and its expression can affect fat deposition. Over-expression of FAS gene can promote cell proliferation, inhibit cell apoptosis, and also affect the synthesis of triglycerides in adipocytes, thereby affecting fat deposition. We have incorporated some of these in the revised manuscript.

L253: Add citation 

Responses from the authors: Thanks for the comment. References are added in L301-303 (i.e., #32-34 in the revised manuscript). 

L262: Is there any refence reporting the meat quality parameter of Zongdihua pigs comparing with other breeds? If so, it could be included to support this hypothesis.

Responses from the authors: Thanks for the excellent comment. Yes, a previous sstudy has shown that the skin thickness of Zongdihua pigs is as high as 0.55cm, which is higher than that of Guanling pigs and Jiangkou radish pigs [x], and the tenderness shear force value is 2.925 kg. Thus, we have added the reference in the revised manuscript.

Feng Wenhao. Study on genetic diversity and genetic relationship of three local pig breeds in Guizhou. Guizhou University, 2008.

Conclusions

As previously mentioned, this affirmation should be excluded or reformulated. Moreover, a perspective of future studies could be included.

Responses from the authors: Thanks for the comment. We have added future plan in the Conclusion section.

Figures

I suggest reporting Figures 2, 5, 6 , and 7 as supplementary. 

Responses from the authors: Thanks for your comments, and we have the revision in the revised manuscript according to your suggestion.

Reviewer #2: 

The title reflects the work's main result which means that instead of “Zongdihua pig tissues” you must use Zongdihua pig isolated adipocytes once the overexpression was done in isolated cells, not in tissue.

Responses from the authors: Thanks for this comment. We have replaced the word “tissues” with “isolated cells”.

ABSTRACT

Line 19. Please, Replace Zongdihua pig by Zongdihua pig isolated adipocyte

Line 19. Replace identify processes by identify biochemical processes

Line 20. Replace, using molecular genetics by molecular tools

Responses from the authors: Thanks for these suggestions. They have been corrected in the revised manuscript.

INTRODUCTION

Line 40. In the first paragraph, please, develop in a deeper way the signaling pathway crosslink between the genes FABP1, FAS, and COL3A1. 

Responses from the authors: Thanks for the excellent point. Per your suggestion, we have added the following into the Introduction section.

The extracellular matrix (ECM) can protect mature adipocytes from being destroyed by external compression and stimulation, and the main component of ECM is collagen [1-2]. Studies have shown that in the process of fat deposition, a large amount of collagen is synthesized [3]. TGF-β positively regulates genes encoding collagen [4], and these collagen-encoding genes are also highly expressed in tissues with higher fat content [5]. In the study of the effect of splicing enzyme (ADAMTS2) in the process of collagen formation on intramuscular fat, it was found that the content of intramuscular fat was significantly positively correlated with the expression of ADAMTS2 gene. It can be seen that there is an intrinsic link between collagen and fat, but the specific regulatory mechanism remains unclear.

Line 53. Please, in the sentence “In the present study… you should increment the paragraph with a clear objective of the study. Which question will be answered?

Responses from the authors: Thanks for the excellent comment. Per your suggestion, we have added some in the Instruction section to clearly lay out the objective. 

MATERIAL AND METHODS

Line 87. Please, add the Accession number (e.g NM_) of the sequences you have used to design oligonucleotides.

Responses from the authors: Thanks for the comment. L83-86 provided FAS, FABP1, GAPDH accession numbers. Additionally, the accession number for LOX gene is NM_001206403 and COL3A1 is NM_001243297, which is also added in the text.

Line 138. If you have used 4 ug of plasmid to transfect, please add this information just before the sentence “recombinant plasmids pEGFP-C1-FABP1 and pEGFP-C1-FAS”. Otherwise, what is the 4ug of DNA? Add empty plasmids controls and blanks to sentence. Clarify this sentence until line 140.

Responses from the authors: Thanks for the comment. Per your suggestion, the text has been revised in the manuscript. 

Line 153. In Data processing and analysis, the 2-△△Ct method is a gene expression method and I do not identify the statistical methods used not only in gene expression analysis, but also in the other analysis. Please, add the proper information and experimental design in this subtopic 

Responses from the authors: Thanks for the comment. 2-△△Ct is the statistical method, and it was used to statistically analyze the experimental data. We have clarified in the text.

RESULTS AND ANALYSIS

Line 163. Please, delete the term “extremely”

Responses from the authors: Thanks for the comment. It has been deleted.

Line184. Delete the term “and differentiate” or changed it to “morphology differentiate” following the idea of the next sentence.

Responses from the authors: Thanks for the comment. The phrase “and differentiate” have been deleted.

Line 191. Don’t you have the quantification the lipids content staining with Oil Red O as you mentioned in MM? Put the data in figure 6 and refer to it here.

Responses from the authors: Thanks for the comment. Oil red O staining was used to identify adipocytes. The prepared oil red O stock solution and distilled water were mixed at a ratio of 3:2 to form an oil red O working solution, and the working solution was added for staining for 1 h. It was not used as a quantitative measurement.

Line 195. Please, the sentence “Cloned versions of FAS and FABP1 were constructed that were fused to the eGFP protein at their carboxyl termini and over-expressed via the strong CMV promoter” is important to MM section. Relocate it. Otherwise, just here the reader will understand what was done.

Responses from the authors: Thanks for the comment. This information has been moved to MM section under 1.6.

FIGURES

Figures 1 and 8. Please, in the Y axis add “/GAPDH” just after relative expression.

Responses from the authors: Thanks for the comment. Your suggestion is accepted and fixed in the figures.

Figures 5 and 6. Please, add magnification bar in the bottom corner of the image Images seems in different magnification.

Responses from the authors: Thanks for your comment. Although it appears different in magnifications, they are indeed the same. We have inserted a magnification bar (100 x) in the bottom of the image in both figures.

Figure 7. Here we have some problems. This figure per se is not a result, but only a control of the efficiency of you transfection. I strongly recommend you to put together this figure to figure 8. In addition, I think we have a problem with the image once only pEGFP-C1-FAS transfected cell have been shown (doubled data). The magnification of the images seems different, please, correct that.

Responses from the authors: The magnifications for Figures 7 and 8 are same. Reviewer one also has comments on those figures. Per your suggestion as well as the other reviewer, we have adjustments in figures to accommodate both of your comments. Thanks again for the excellent comments. 

Reviewer #3: 

The manuscript entitled “Fatty acid binding protein 1 and fatty acid synthetase overexpression have differential effects on collagen III and cross-linking in Zongdihua pig tissues” aims to determine whether FABP1 and FAS regulate expression of collagen and its crosslinking via lysyl oxidase in Zongdihua pigs. The manuscript points out that enhancing the expression of FAS and FABP1 increases collagen accumulation and this preliminarily suggests that FAS and FABP1 can serve as fat-related candidate genes providing a theoretical basis for the study of fat deposition in Zongdihua pigs. The manuscript brings interesting results, however, to be accepted for publication, it needs major changes and corrections. Below are suggestions to make the manuscript suitable for publication and more informative to the reader.

Responses from the authors: Thanks for the comments. We are committed to work with you to improve the manuscript.

General suggestions

- Write all gene names in italics. Example: FABP1

Responses from the authors: Thanks for the comment. We have checked and corrected them.

- Please review the text for language.

Responses from the authors: The revised manuscript has been proofread by a native English speaker.

- Some words are used in different ways in the manuscript such as cross-linking/crosslinking, Over-expression/Overexpression…Correct this, please.

Responses from the authors: Thanks for the comment. The word has been corrected. Also, the revised manuscript has been proofread by a native English speaker.

- The introduction needs to be rewritten, addressing the issues related to the manuscript more deeply. In addition, give a clear hypothesis and objective in the introduction section.

Responses from the authors: Thanks for your comment. We have added some regarding the pig breed and why it is a good candidate. 

- Please, change “longissimus dorsi” to “Longissimus dorsi” throughout the manuscript.

Responses from the authors: Thanks, and we have changed them accordingly. 

- Discussion needs to be rewritten, addressing more deeply the observed results.

Responses from the authors: Thanks for your comments. Per your suggestion, we have revised some parts in the Discussion section. 

Abstract

- Line 26 - Remove the with in the sentence "FAS encodes 333 amino acid hydrophobic protein containing with 26 phosphorylation sites and 0 transmembrane regions".

Responses from the authors: Thanks for the comment. It is corrected in the revised manuscript.

- Line 27 - Change the following sentence "The basal levels of FABP1 and FAS in pig tissues expression were 3 -3.5-fold higher in subcutaneous fat compared with muscle (P < 0.01)" to "The basal levels of FABP1 and FAS expression in pig tissues were 3 -3.5-fold higher in subcutaneous fat compared with muscle (P < 0.01)".

Responses from the authors: Thanks for the comment. The sentence is corrected. 

- Line 33- I believe you forgot to add “FAS” in this sentence "Therefore, enhancing FABP1 expression increases collagen accumulation...". Am I correct? Because the overexpression of FAS and FABP1 resulted in a significantly increased expression of the COL3A1 gene.

Responses from the authors: Thanks for the comments. Yes, you are correct, and the missing word “FAS” has been added.

- Keywords must be arranged in alphabetical order. It is not recommended to add words that are already in the title of the manuscript such as Zongdihua pig.

Responses from the authors: Thanks. They are corrected per your suggestion.

Introduction

- Add the title “Introduction” before the text in the line 40.

Responses from the authors: Yes, it is added.

- You could improve your introduction by talking more about the breed (Zongdihua pig), mainly about the meat traits. 

Responses from the authors: Thanks for your comment. We have added the following in the Introduction section, which reads: 

Zongdi Huazhu is mainly produced in Zongdi Town, Guizhou Province. It is one of the local pig breeds in Guizhou with excellent meat quality. It has good motherhood, strong adaptability and disease resistance. The meat is fragrant and glutinous, with the meat quality characteristics of high-grade meat products. Zongdihua pigs are fatty pigs with a lean meat rate of only 38%. Studies have shown that Zongdi Hua pork has better quality properties than large white pigs [8]. 

[8] Zhang Yun, Chen Wei, Li Ping, et al. Carcass and meat quality traits of land-colored pigs with different rearing methods. Guizhou Agricultural Science, 2014, 42(9):3.

It could also present some preliminary studies that address the evaluated genes deeply. All these issues need to be better integrated. In addition, the objective of the work should be better presented in this introduction.

Responses from the authors: Thanks for the comments. No, sorry that we did not have additional study data to add. Due to strict Covid-19 control policy in China, adding new lab studies at this time is extremely difficult. However, we will keep this in mind and will incorporate the suggestion to our future studies.

Materials and methods

- Line 58 - Correct to Ethics Statements.

Responses from the authors: Thanks. It is fixed.

- Why is the topic "Ethics Statements" highlighted in the text?

Responses from the authors: We have removed the highlight.

- Please, I ask you to correct the text in relation to the language used. Some sentences are poorly worded.

Responses from the authors: Thanks for the comments. We have our revised manuscript checked by a native English speaker. 

- You don't write about the design and accession number of the following genes: LOX and COL3A1 in the text.

Responses from the authors: Thanks for the comments. Per suggestion, we have added the information, i.e., LOX gene: NM_001206403; COL3A1: NM_001243297

- Did you work with the fold change (comparative CT method)? Why are the graphs with the Y-axis legend as a relative expression? I suggest reading the article "Analyzing real-time PCR data by the comparative CT method - Thomas D Schmittgen1 & Kenneth J Livak 2008" to correct this question

Responses from the authors: Thanks for the comment. The method used in this study was to present relative gene expression; thus, it is a comparative C(T) method. The Y-axis of the graph represents the calculated relative expression of the gene. 

- Subcutaneous fat and Longissimus dorsi samples were collected from how many animals? 

Responses from the authors: Thanks for your comment. Subcutaneous fat and muscle tissues were collected from 6 9-month-old pigs

What is the N of this gene expression analysis? You need to better describe the experimental design of the study.

Responses from the authors: Thanks for the comment. Not quite clear about the comment. Is this regarding mRNA expression analysis? Based on other comments, we have added some in the section.

Results

- Why in the legend of figure 1 you used fat deposition if there is only information about the fold change between subcutaneous fat and Longissimus dorsi? 

Responses from the authors: Thanks for the comment. Fat deposition in Figure 1 does not refer to tissue, the legend means more about the expression of fat deposition-related genes (FABP1 and FAS) and collagen-related genes (LOX or COL3A1) in porcine subcutaneous fat and muscle tissue.

- Line 163:164 – “The expression levels for all 4 genes were extremely significantly (P < 0.01) elevated in subcutaneous fat tissues compared with the muscle (control) tissues (Figure 1).” Why did you compare subcutaneous fat to muscle? 

Responses from the authors: Thanks for the comment. Zongdihua pigs are fat-rich pigs, they have high fat content with thick back fat and thick skin. Also, because their meat is chewy and full of glutinous aroma, their meat quality is generally considered good. Therefore, in order to explore the characteristics of Zongdihua pigs, the subcutaneous fat and muscle are selected tissue targets for comparison.

Your hypothesis should be very clear in the manuscript in relation to the comparisons made. Besides that, could you explain a little more about what you considered as a control in this study? 

Responses from the authors: Thanks for your comment. In this study, FABP1 and FAS genes were highly expressed in adipocytes, which helps verifying whether high expression of FABP1 and FAS genes would affect the expression of collagen-related genes, and it further helps exploring the mechanism of fat deposition on collagen and cross-linking.

- Is the Y-axis legend correct in Figure 9?

Responses from the authors: Thanks for the comment. Yes, we have double checked and it is correct.

Discussion

- Line 225:226 – Change the sentence “In this study, we found that FABP1 and FAS gene expression were significantly higher than for the control muscle tissue” to “In this study, we found that FABP1 and FAS gene expression were significantly higher in subcutaneous fat than for the control muscle tissue”

Responses from the authors: Thanks for the comment, and we have replaced the sentence per your suggestion.

- Line 232 – Alter LOC gene to LOX.

Responses from the authors: Thanks, and this is fixed. We also, searched the entire manuscript to make sure that there is no LOC.

- Your discussion of the results is superficial, you need to go deeper into the effect of the FAS and FABP1 genes on the expression of COL3A1 and LOX and the production of collagen. In addition to the effects of these genes on the deposition of fat and on meat traits.

Responses from the authors:

- Line 231:233- “Our results agreed with previous studies indicating that COL3A1 and LOC gene expression in subcutaneous fat are significantly higher and lower than in longissimus dorsi muscle tissues, respectively.” - Figure 1 shows greater expression of both the COL3A1 and LOX genes in subcutaneous fat in relation to Longissimus dorsi.

Responses from the authors: Thanks for the comments. It was our mistake. Now, the sentence in the text only addresses COL3A1, so it matches with the result showing in Figure 1.

- You have not discussed deeply the effects of the expression of these genes on meat quality in the breed studied.

Responses from the authors: Thanks for your comment! Per your suggestion, we have added the following in the Discussion section. 

After overexpression of FAS and FABP1 genes, the expression of COL3A1 increased and the expression of LOX decreased, indicating that with the deposition of fat, collagen synthesis increased, while the expression of LOX was inhibited, resulting in a decrease in mature cross-linking content. Therefore, the high expression of these genes makes Zongdihua pigs high in fat and collagen, but low in cross-linking, resulting in thick and waxy skin and better meat tenderness.

---

## [Decision Letter · Decision Letter 1]

17 Oct 2022

PONE-D-22-15349R1Fatty acid binding protein 1 and fatty acid synthetase over-expression have differential effects on collagen III synthesis and cross-linking in Zongdihua pig primary adipocytesPLOS ONE

Dear Dr. Zhou,

Thank you for submitting your manuscript to PLOS ONE. After careful consideration, we feel that it has merit but does not fully meet PLOS ONE’s publication criteria as it currently stands. Therefore, we invite you to submit a revised version of the manuscript that addresses the points raised during the review process.

Please submit your revised manuscript by Dec 01 2022 11:59PM If you will need more time than this to complete your revisions, please reply to this message or contact the journal office at plosone@plos.org. Please include the following items when submitting your revised manuscript:A rebuttal letter that responds to each point raised by the academic editor and reviewer(s). You should upload this letter as a separate file labeled 'Response to Reviewers'.A marked-up copy of your manuscript that highlights changes made to the original version. You should upload this as a separate file labeled 'Revised Manuscript with Track Changes'.An unmarked version of your revised paper without tracked changes. You should upload this as a separate file labeled 'Manuscript'.

We look forward to receiving your revised manuscript.

Kind regards,

Marcio Duarte, PhD

Academic Editor

PLOS ONE

Reviewers' comments:

Reviewer's Responses to Questions

**Comments to the Author**

1. If the authors have adequately addressed your comments raised in a previous round of review and you feel that this manuscript is now acceptable for publication, you may indicate that here to bypass the “Comments to the Author” section, enter your conflict of interest statement in the “Confidential to Editor” section, and submit your "Accept" recommendation.

Reviewer #1: (No Response)

Reviewer #2: (No Response)

2. Is the manuscript technically sound, and do the data support the conclusions?

Reviewer #1: Yes

Reviewer #2: Yes

3. Has the statistical analysis been performed appropriately and rigorously? 

Reviewer #1: I Don't Know

Reviewer #2: I Don't Know

4. Have the authors made all data underlying the findings in their manuscript fully available?

Reviewer #1: Yes

Reviewer #2: Yes

5. Is the manuscript presented in an intelligible fashion and written in standard English?

Reviewer #1: Yes

Reviewer #2: Yes

6. Review Comments to the Author

Reviewer #1: Title: Fatty acid binding protein 1 and fatty acid synthetase over-expression have differential effects on collagen III synthesis and cross-linking in Zongdihua pig primary adipocytes

I have noticed some changes in the current manuscript version, however, some suggestions and questions were answered but not addressed in the manuscript. The general introduction and discussion need to be restructured to provide a better link between the paragraphs and a better understanding of the reader. Please consider the suggestions bellow:

Abstract

L30-31: “We aimed…”

L32: breeding improvement in terms of what?

L37: Please remove “FAS encodes …….regions.”

Introduction

L52- 70: These two paragraphs need extensive revision mainly in term of sentence structure. For example, L52-55 can be transformed into 3 sentences “The extracellular matrix (ECM) play roles in adipocyte proliferation, differentiation, and migration. Moreover, ECM anchors these cells to prevent mechanical movement and protect mature adipocytes from external compression and stimulation. It is well established that collagen is the main component of ECM ….”

L81-85 This paragraph is disconnected from the rest of the text, please reorganize the introduction section. I would suggest to focus and organize this section by following the topics: Breed and meat characteristics, ECM and its components, FABP1 and it regulatory role, and a clear description of the objectives (The present study aimed to investigate x, y and z …)

Material and Methods

L95: “Healthy Zongdihua pigs including …”

-Section 1.3 (Reagents) and 1.4 (primer design and synthesis): the information in these sections should be placed with its respective analyses and these topics removed, as I have already mentioned in the first revision.

- The accession numbers described in section 1.4 could be better visualized if placed in a column of Table 1.

L125: The total RNA was extracted from fat and longissimus dorsi muscle from pigs in which age? 3d , 9 mo or both? Since you have collected tissue from both ages and in some analysis you used only one age, make sure you are being specific throughout the manuscript.

L155: by DNA sequencing? How was done the bioinformatic analysis of DNA sequencing data? Please add a description or reference of the employed methodology.

L187: It is not clear what was considered in the statistical model, fixed and random effects. All the information needs to be clear to allow reproducibility.

Results

L192: “2 Results”

L193: “2.1 Differential expression analysis”

L164: please specify the age of the pigs in text and Figures.

L236: “… were examined for lysyl …”

L239-240: “(Figure 5B)”

Discussion

L260-261: This is a strong affirmation. Since you compared two different cell lines (from muscle vs adipose tissue), it is expected, and already reported, greater influence of this genes on adipocytes compared to muscle. To ensure that these genes are associated with greater fat deposition ability you should have evaluated pigs with different levels of subcutaneous fat, different breeds, etc. Please rewrite the indicative of this data or provide references of the influence of these genes on greater subcutaneous fat deposition.

L274-277: repeated sentences.

Similar to introduction, the paragraphs seem disconnected. Please improve the link between paragraphs for better understanding and clarity.

Conclusion: ok

Figure 3. Please verify the letters.

Figure 4 “…. COL3A1..” ; transfected with (A) pEGFP-C1-FAS and (B) pEGFP-C1-FAS? Please correct figure’s legend.

Reviewer #2: Dear Authors,

Thank you to considering most of questions raised previously to make the manuscript suitable for publication and more informative to the reader.

Most of concerns were properly addressed, but I still have some serius concerns about the Statistics.

To make the data more informative, at the Data processing topic, 2^-ddCt is a method to access the fold relative gene expression. In order the access the differences in gene expression between the groups you must use a proper statistical model and test (parametric or non-parametric) and I still have not seen it in the text. Please, it is important to add not only the software, but also the test you have used. In addition, please add a supplementary table at the supplementary material with cT collected data from qPCR for each replicate, dissociation curves for each gene reaction and the statistical metrics.

7. PLOS authors have the option to publish the peer review history of their article (what does this mean?). If published, this will include your full peer review and any attached files.

Reviewer #1: No

Reviewer #2: No

---

## [Author Response · Author response to Decision Letter 1]

2 Dec 2022

Dear Editors and Reviewers:

Thank you for your letter and for the reviewers’ comments concerning our manuscript entitled“Fatty acid binding protein 1 and fatty acid synthetase over-expression have differential effects on collagen III synthesis and cross-linking in Zongdihua pig primary adipocytes”(ID:PONE-D-22-15349R1). Those comments are all valuable and very helpful for revising and improving our paper, as well as the important guiding significance to our researches. We have studied comments carefully and have made correction which we hope meet with approval. Revised portion are marked in yellow on the revised version. The main corrections in the paper and the responds to the reviewer’s comments are as flowing:

Responds to the reviewer’s comments:

Reviewer #1: 

Title: Fatty acid binding protein 1 and fatty acid synthetase over-expression have differential effects on collagen III synthesis and cross-linking in Zongdihua pig primary adipocytes.I have noticed some changes in the current manuscript version, however, some suggestions and questions were answered but not addressed in the manuscript. The general introduction and discussion need to be restructured to provide a better link between the paragraphs and a better understanding of the reader. Please consider the suggestions bellow:

Abstract

1.L30-31: “We aimed…”

Responses from the authors: Thanks for your excellent suggestions.Modified according to your suggestions.

2.L32: breeding improvement in terms of what?

Responses from the authors: Through the research on the meat quality characteristics of the Landflower pig, the dominant traits of the breed can be excavated, and the traits such as slow growth, small size and low reproduction rate can be improved while retaining these excellent traits in the future breeding process.

3.L37: Please remove “FAS encodes …….regions.”

Responses from the authors: Thanks for this comment.We have deleted as required.

Introduction

4.L52- 70: These two paragraphs need extensive revision mainly in term of sentence structure. For example, L52-55 can be transformed into 3 sentences “The extracellular matrix (ECM) play roles in adipocyte proliferation, differentiation, and migration. Moreover, ECM anchors these cells to prevent mechanical movement and protect mature adipocytes from external compression and stimulation. It is well established that collagen is the main component of ECM ….”

Responses from the authors: Thanks for your excellent suggestions.Your points are well taken. Per your suggestion, in the revised manuscript,We revised the sentence structure.L52-55 is modified as follows:The extracellular matrix (ECM) play roles in adipocyte proliferation, differentiation, and migration. Moreover, ECM anchors these cells to prevent mechanical movement and protect mature adipocytes from external compression and stimulation. It is well established that collagen is the main component of ECM

5.L81-85 This paragraph is disconnected from the rest of the text, please reorganize the introduction section. I would suggest to focus and organize this section by following the topics: Breed and meat characteristics, ECM and its components, FABP1 and it regulatory role, and a clear description of the objectives (The present study aimed to investigate x, y and z …)

Responses from the authors: Thanks for your excellent summary comments. Your points are well taken. Per your suggestion,We have modified and supplemented this part.

L81-85 The revised contents are as follows:Zongdihua pig is mainly produced in Zongdi Township, Ziyun County, Guizhou Province. It has strong disease resistance, tolerance to roughage, and a strong ability to digest and absorb crude fiber. Its meat features excellent quality, bright red flesh color, rich intramuscular fat, high water tying rate, little cooking loss, thick carcass skin, soft and waxy texture, easy to cook, and tender and fragrant muscles, which is highly appreciated by consumers. It is a valuable local genetic resource with a high value of development and utilization [11, 16]. As mentioned above, collagen is the main component of ECM, which has an important influence on the tenderness, viscoelasticity and chewiness of the meat, and it will be synthesized in large quantities when fat is deposited. However, Zongdihua pig is a fatty breed, and its lean meat rate is only 38%. In addition, the study shows that the polymorphism of the FABP1 gene is significantly related to the fat content of chest muscle, which is expressed in various tissues of pigs. It is speculated that the FABP1 gene may be the main gene affecting the intramuscular fat content of pigs, or a marker gene closely related to the main gene. Therefore,the present study aimed to investigate the effects of FABP1 and FAS on the cross-linking of collagen and pyridine, and provide a molecular theoretical basis for the development and utilization of local pig genetic resources.

Material and Methods

6.L95: “Healthy Zongdihua pigs including …”

Responses from the authors: Thanks for this comment.Healthy Zongdihua pigs including conform to the characteristics of this breed, have good growth performance and have no diseases.

7.Section 1.3 (Reagents) and 1.4 (primer design and synthesis): the information in these sections should be placed with its respective analyses and these topics removed, as I have already mentioned in the first revision.The accession numbers described in section 1.4 could be better visualized if placed in a column of Table 1.

Responses from the authors: Thanks for this suggestion. Per your comment, it has been changed according.The information in 1.3 (Agents) and 1.4 (print design and synthesis) has been put together with the corresponding analysis, and these topics have been deleted. At the same time, in order to be more intuitive, the accession number described in Section 1.4 has been placed in Table 1.(Red font)

Table 1. PCR primers used in this study

Gene Numbers Sequence (5’→3’） Annealing temperature/℃ Product length/bp Purpose

FABP1 NM_001004046.2 F:TTGGGAGAGGAGTGTGAGATG 

R:CACAGACTTGATGCCTTTGAA 58 111 qPCR

FAS NM_213839.1 F:CCTGTATCGCTGGACCACT

R:GGGCACTCAGACTCCCTTT 57.9 103 qPCR

GAPDH XM_021091114.1 F:TTGTGATGGGCGTGAACC

R:GTCTTCTGGGTGGCAGTGAT 60.0 169 qPCR

LOX NM_001206403 F:TACCAAGCCGACCAAGATA '

R:GTGTGCAGTACAGGCAAAT 57.2 147 qPCR

COL3A1 NM_001243297 F:TTTGCTCTACTTCATCCCACT 

R:CTTCCAGACATCTCTATCCGC 56.5 105 qPCR

FABP1-CDS NM_001004046.2 F: CCGATGAACTTCTCCGGCAAATACC

R:CACGGTACCCTCTTGCTGATTCTCTTGAAGACAA 58.5 384 Cloning

FAS-CDS NM_213839.1 F:CCGATGTCCGGGATCTGGGTTCT 

R:CACGGTACCCTAGGTCAAGCTTTCATTCTCATTC 60.0 999 Cloning

8.L125: The total RNA was extracted from fat and longissimus dorsi muscle from pigs in which age? 3d , 9 mo or both? Since you have collected tissue from both ages and in some analysis you used only one age, make sure you are being specific throughout the manuscript.

Responses from the authors: Thanks for the comment.Extraction of total RNA from fat and longissimus dorsi muscle of 9 month old pigs. This information has been edited in the revised manuscript.

9.L155: by DNA sequencing? How was done the bioinformatic analysis of DNA sequencing data? Please add a description or reference of the employed methodology.

Responses from the authors: Thanks for the comment.A comparison of the DNA sequencing results in NCBI showed that the results were consistent with the original sequence. The physicochemical properties, secondary structures, transmembrane regions, and phosphorylation sites of FABP1 and FAS proteins are analyzed by the bioinformatics analysis software.The method adopted in the revised manuscript has been supplemented.

10.L187: It is not clear what was considered in the statistical model, fixed and random effects. All the information needs to be clear to allow reproducibility.

Responses from the authors: Thanks for this comment.Fixed effect model is considered to be used as the statistical model, which aims to compare the expression of related genes in different tissues in this experiment. These comparison objects are fixed, not randomly selected. We use a one-way analysis of variance (F-test) to choose the form of the data model.

Results

11.L192: “2 Results”

Responses from the authors: Thanks for this suggestion. Per your comment, it has been changed according.

12.L193: “2.1 Differential expression analysis”

Responses from the authors: Thanks for this suggestion. Per your comment, it has been changed according.

13.L164: please specify the age of the pigs in text and Figures.

Responses from the authors: Thanks for the comment. We indicated the age of pigs in the revised manuscript.

14.L236: “… were examined for lysyl …”

Responses from the authors: Per suggestion, it has been changed according.

15.L239-240: “(Figure 5B)”

Responses from the authors: Per your comment, it has been changed according.

Discussion

16.L260-261: This is a strong affirmation. Since you compared two different cell lines (from muscle vs adipose tissue), it is expected, and already reported, greater influence of this genes on adipocytes compared to muscle. To ensure that these genes are associated with greater fat deposition ability you should have evaluated pigs with different levels of subcutaneous fat, different breeds, etc. Please rewrite the indicative of this data or provide references of the influence of these genes on greater subcutaneous fat deposition.

Responses from the authors: Thanks for your excellent comment. Per your suggestion, we added the following in the Discussion section, which reads:

At the same time，alterations in the primary sequences that would affect enzymatic functions were not detected.The expression level of FABP1 and FAS in muscle and adipose tissue is different among breeds and within breeds. The expression level of subcutaneous adipose tissue in Kele pig and DLY crossbred pig is significantly higher than that in longissimus dorsi muscle, and the expression level of subcutaneous adipose FABP1 in different breeds is also different. The expression level of subcutaneous adipose FABP1 in the Kele pig is higher than that in the DLY crossbred pig, which may be due to the fact that the Kele pig is a local pig of a fatty breed with large fat deposition. In this study, the expression of FABP1 and FAS genes in the subcutaneous adipose tissue of Zongdihua pig is significantly higher than that of the longissimus dorsi muscle, which is similar to other pig breeds.

17.L274-277: repeated sentences.Similar to introduction, the paragraphs seem disconnected. Please improve the link between paragraphs for better understanding and clarity.

Responses from the authors: Good comment.Per your suggestion,we have changed it in the text:

The studies show that the expression of the COL3A1 gene in pig subcutaneous fat is significantly higher than that in longissimus dorsi muscle. Results were consistent with higher levels of collagen in fat compared with muscle tissues. The expression level of the LOX gene is lower than that in longissimus dorsi muscle, and the content of pyridine cross-linking (HP and LP) is the lowest in subcutaneous fat. We speculate that this might be related to fat deposition. We therefore cultured primary adipocytes from subcutaneous fat tissues in these pigs. We were able to successfully over-express FAS and FABP1 in Zongdihua pig preadipocytes and this resulted in increased COL3A1 and decreased LOX expression levels (P<0.01) as well as lysyl oxidase activities. These results indicate that fat deposition may disrupt the structure of connective tissues resulting in an increase in tenderness with age. 

18.Conclusion: ok

Responses from the authors：Thanks!We look forward to your "acceptance" suggestions and responses！

19.Figure 3. Please verify the letters.

Responses from the authors: Thanks for this comment. They are checked.

20.Figure 4 “…. COL3A1..” ; transfected with (A) pEGFP-C1-FAS and (B) pEGFP-C1-FAS? Please correct figure’s legend.

Responses from the authors: Thanks for this comment.It is modified in the figure legend:transfected with (A) pEGFP-C1-FABP1 and (B) pEGFP-C1-FAS. 

Reviewer #2: 

Dear Authors,

Thank you to considering most of questions raised previously to make the manuscript suitable for publication and more informative to the reader.Most of concerns were properly addressed, but I still have some serius concerns about the Statistics.To make the data more informative, at the Data processing topic, 2^-ddCt is a method to access the fold relative gene expression. In order the access the differences in gene expression between the groups you must use a proper statistical model and test (parametric or non-parametric) and I still have not seen it in the text. Please, it is important to add not only the software, but also the test you have used. In addition, please add a supplementary table at the supplementary material with cT collected data from qPCR for each replicate, dissociation curves for each gene reaction and the statistical metrics.

Responses from the authors:Thanks for your excellent comments.The 2-△△Ct method also referred to as the comparative C T method.This method is used to calculate the relative quantification of target gene and reference gene, and compare with reference gene.Mathematical model for relative quantification in real- time PCR

The mathematical statistical model of relative quantification in real-time PCR of this test is calculated according to the method provided by Schmittgen T D and Michael

[1] Schmittgen T D . Analyzing real-time PCR data by the comparative CT method[J]. Nature Protocols, 2008.

[2]Michael, W, Pfaffl. A new mathematical model for relative quantification in real-time RT–PCR[J]. Nucl Acids Res, 2001.

In addition, add a table about your proposal, including each duplicate ct data collected from qPCR. Due to the outbreak of the COVID-19 in China, we cannot retrieve the computed CT data, but we have retained the dissociation curve and statistical indicators of each gene reaction in qPCR. We have added this part to the "Fig. S5 The sampling curve and measuring curve of (A) COL3A1, (B) LOX, (C) FAS, (D) FABP1, (E) GAPDH gene" in the supplementary materials

---

## [Decision Letter · Decision Letter 2]

20 Feb 2023

PONE-D-22-15349R2Fatty acid binding protein 1 and fatty acid synthetase over-expression have differential effects on collagen III synthesis and cross-linking in Zongdihua pig primary adipocytesPLOS ONE

Dear Dr. Zhou,

Thank you for submitting your manuscript to PLOS ONE. After careful consideration, we feel that it has merit but does not fully meet PLOS ONE’s publication criteria as it currently stands. Therefore, we invite you to submit a revised version of the manuscript that addresses the points raised during the review process.

We look forward to receiving your revised manuscript.

Kind regards,

Marcio Duarte, PhD

Academic Editor

PLOS ONE

Journal Requirements:

Reviewers' comments:

Reviewer's Responses to Questions

**Comments to the Author**

1. If the authors have adequately addressed your comments raised in a previous round of review and you feel that this manuscript is now acceptable for publication, you may indicate that here to bypass the “Comments to the Author” section, enter your conflict of interest statement in the “Confidential to Editor” section, and submit your "Accept" recommendation.

Reviewer #1: All comments have been addressed

Reviewer #2: (No Response)

2. Is the manuscript technically sound, and do the data support the conclusions?

Reviewer #1: Yes

Reviewer #2: Partly

3. Has the statistical analysis been performed appropriately and rigorously? 

Reviewer #1: I Don't Know

Reviewer #2: No

4. Have the authors made all data underlying the findings in their manuscript fully available?

Reviewer #1: Yes

Reviewer #2: No

5. Is the manuscript presented in an intelligible fashion and written in standard English?

Reviewer #1: Yes

Reviewer #2: Yes

6. Review Comments to the Author

Reviewer #1: The answers were addressed, and I could realize an improvement throughout the text. However, it still need some minor revision, mainly regarding the clarity in some sentences to make the paper more objective to the reader. Please consider the comments below.

Abstract: ok

Introduction

L48-49: “…fibrillar collagens and the expression….”

L55: “…such as TGF-β, which is a positive…”

L56: “…collagen genes and it is highly…”

L71-85: I suggest to remove this paragraph

Material and Methods

L95-96: not necessary to report “( conform to the characteristics of this breed, have good growth

performance and have no diseases)”. I suggest to remove

L135: “…proteins were analyzed…”

L144: Please remove “as per above” here and throughout the text

L145: “The fat cell of 3-day-old pigs were cultured and the pellets were suspended…”

L162: please correct “37oC”

Results

L201: Please verify the Figure’s citation here and throughout the text

L224: Figure 4 shows the P < 0.01 for COL3A1 expression in both over-expressed clones. Please correct this information here or in the Figure

L226-227: Please correct this sentence for “We further examined the effects of transfected plasmids on LOX mRNA levels and lysyl oxidase activity.”

L229: “..double-antibody sandwich ELISA method…”

Discussion

L253-256: Please substitute the sentence to: “The expression level of subcutaneous adipose tissue in Kele pig and DLY crossbred pig is significantly higher than longissimus dorsi muscle” and add a citation.

L256-258: Please substitute the sentence to: “Moreover, the expression level of subcutaneous adipose FABP1 in the Kele pig is higher than DLY crossbred pig, which may be due to the fact that the Kele pig is a local pig that present a high fat deposition” and add a citation.

L259: “…significantly higher compared to longissimus dorsi muscle…”

L260: “These data may indicate that FABP1….”

L263: which studies? Please cite

L264: “significantly higher compared to longissimus dorsi muscle. The results…”

L265: “….compared to muscle”

L265-267: The expression of LOX is lower in fat compared to muscle or the expression of LOX is lower in muscle compared to fat?? This sentence is confused. Please rephrase this sentence and cite the paper.

L267-270: I suggest to substitute by “By culturing primary adipocyte from subcutaneous fat we were able

to successfully over-express FAS and FABP1 in Zongdihua pig preadipocytes and this resulted in increased COL3A1 and decreased LOX expression levels (P<0.01) as well as lysyl oxidase activities”

L277: “…gene products collectively affected…”

L290: “…while inhibited LOX…”

Conclusion: COL3A1 is in italic, but the other genes are not, please follow a pattern throughout the text. I suggest reporting mRNAS in italic.

Supplementary material:

Table S1, S2 and S3: Unnecessary to report. I suggest to remove.

Figure S5 should be cited in the text

Reviewer #2: Dear author,

I got your point and the comparative dct method seems correct, but I still have some serious concerns about the statistics you should have done after that.

At the Data processing and statistical topic, 2^-ddCt is a method to access the fold relative gene expression, which means that using this mathematical model you are able to compare different groups in terms of gene expression level e.g. how many times a gene is more expressed in one group than another. In another hand, in order to access if there is a statistical difference between groups you must use a proper statistical model/test (parametric or non-parametric) after ddct method; and I still have not seen it in the text. A one-way ANOVA or t-test would be enough for you. Once the statistical metrics have not been properly addressed I can not trust any qPCR results.

I strongly suggest you to perform the statistical test and potential corrections and resubmit the manuscript.

7. PLOS authors have the option to publish the peer review history of their article (what does this mean?). If published, this will include your full peer review and any attached files.

Reviewer #1: No

Reviewer #2: No

---

## [Author Response · Author response to Decision Letter 2]

9 Mar 2023

Dear Editors and Reviewers:

 Thank you for your letter and for the reviewers’ comments concerning our manuscript entitled“Fatty acid binding protein 1 and fatty acid synthetase over-expression have differential effects on collagen III synthesis and cross-linking in Zongdihua pig primary adipocytes”(ID:PONE-D-22-15349R2). Those comments are all valuable and very helpful for revising and improving our paper, as well as the important guiding significance to our researches. We have studied comments carefully and have made correction which we hope meet with approval. Revised portion are marked in yellow on the revised version. The main corrections in the paper and the responds to the reviewer’s comments are as flowing:

Responds to the reviewer’s comments:

Reviewer #1: 

The answers were addressed, and I could realize an improvement throughout the text. However, it still need some minor revision, mainly regarding the clarity in some sentences to make the paper more objective to the reader. Please consider the comments below.

Abstract: ok

Responses from the authors：Thanks!We look forward to your "acceptance" suggestions and responses！

Introduction

L48-49: “…fibrillar collagens and the expression….”

Responses from the authors: Thanks for your excellent suggestions.Modified according to your suggestions.

L55: “…such as TGF-β, which is a positive…”

Responses from the authors: Thanks for your excellent suggestions.Modified according to your suggestions.

 L56: “…collagen genes and it is highly…”

Responses from the authors: Thanks for your excellent suggestions.Modified according to your suggestions.

 L71-85: I suggest to remove this paragraph

Responses from the authors: Thanks for this suggestion. Per your comment, it has been changed according

Material and Methods

 L95-96: not necessary to report “( conform to the characteristics of this breed, have good growth performance and have no diseases)”. I suggest to remove

Responses from the authors: Thanks for your excellent suggestions.We have removed it in the revised version.

 L135: “…proteins were analyzed…”

Responses from the authors: Per suggestion, it has been changed according.

L144: Please remove “as per above” here and throughout the text

Responses from the authors: Thanks for your excellent suggestions.We have removed it in the revised version.

 L145: “The fat cell of 3-day-old pigs were cultured and the pellets were suspended…”

Responses from the authors: Thanks for your excellent suggestions.Modified according to your suggestions.

 L162: please correct “37oC”

Responses from the authors: Per suggestion, it has been corrected according.

Results

 L201: Please verify the Figure’s citation here and throughout the text

Responses from the authors: Thanks for your excellent suggestions. Per your suggestion,We have modified and verified this part.

 L224: Figure 4 shows the P < 0.01 for COL3A1 expression in both over-expressed clones. Please correct this information here or in the Figure

Responses from the authors: Thanks for your excellent summary comments. Your points are well taken. Per your suggestion,We have modified and verified this part.After repeated verification, we have corrected Figure 4 and made correct modifications in the text.

L226-227: Please correct this sentence for “We further examined the effects of transfected plasmids on LOX mRNA levels and lysyl oxidase activity.”

 Responses from the authors: Per suggestion, it has been corrected according.

 L229: “..double-antibody sandwich ELISA method…”

 Responses from the authors: Thanks for this suggestion. Per your comment, it has been changed according.

Discussion

L253-256: Please substitute the sentence to: “The expression level of subcutaneous adipose tissue in Kele pig and DLY crossbred pig is significantly higher than longissimus dorsi muscle” and add a citation.

Responses from the authors: Thanks for your excellent summary comments. Per your suggestion,We have modified this part.Dear Reviewer,Because of the sentence It is the result of this experiment, so no citation is added.

L256-258: Please substitute the sentence to: “Moreover, the expression level of subcutaneous adipose FABP1 in the Kele pig is higher than DLY crossbred pig, which may be due to the fact that the Kele pig is a local pig that present a high fat deposition” and add a citation.

Responses from the authors: Thanks for your excellent suggestions.Modified according to your suggestions.

L259: “…significantly higher compared to longissimus dorsi muscle…”

Responses from the authors: Thanks for your excellent suggestions.Modified according to your suggestions.

L260: “These data may indicate that FABP1….”

Responses from the authors: Thanks for your excellent suggestions.Modified according to your suggestions.

L263: which studies? Please cite

Responses from the authors: Thanks for this comment.The study here refers to the author's previous research results, citation 29

L264: “significantly higher compared to longissimus dorsi muscle. The results…”

Responses from the authors: Thanks for your excellent suggestions.Modified according to your suggestions.

L265: “….compared to muscle”

Responses from the authors: Thanks for your excellent suggestions.Modified according to your suggestions.

L265-267: The expression of LOX is lower in fat compared to muscle or the expression of LOX is lower in muscle compared to fat?? This sentence is confused. Please rephrase this sentence and cite the paper.

Responses from the authors: Thanks for your excellent comment.We have modified this sentence.

The expression of LOX is lower in fat compared to muscle, and the content of pyridine cross-linking (HP and LP) is the lowest in subcutaneous fat[29]. 

L267-270: I suggest to substitute by “By culturing primary adipocyte from subcutaneous fat we were able to successfully over-express FAS and FABP1 in Zongdihua pig preadipocytes and this resulted in increased COL3A1 and decreased LOX expression levels (P<0.01) as well as lysyl oxidase activities”

Responses from the authors: Thanks for your excellent suggestions.We have substituted this part.

L277: “…gene products collectively affected…”

Responses from the authors: Per suggestion, it has been changed according.

L290: “…while inhibited LOX…”

Responses from the authors: Per suggestion, it has been changed according.

Conclusion

COL3A1 is in italic, but the other genes are not, please follow a pattern throughout the text. I suggest reporting mRNAS in italic.

Responses from the authors: Thanks for your excellent comment.A pattern has been followed ithroughout the text.According to your suggestion, use italics for all genes and mRNA

Supplementary material:

Table S1, S2 and S3: Unnecessary to report. I suggest to remove.

Responses from the authors: Thanks for this comment. They are removed.

Figure S5 should be cited in the text

Responses from the authors: Thanks for your excellent suggestions.Reference to Figure S5 has been added to the article(L95).

Reviewer #2:

I got your point and the comparative dct method seems correct, but I still have some serious concerns about the statistics you should have done after that.At the Data processing and statistical topic, 2-△△Ct is a method to access the fold relative gene expression, which means that using this mathematical model you are able to compare different groups in terms of gene expression level e.g. how many times a gene is more expressed in one group than another. In another hand, in order to access if there is a statistical difference between groups you must use a proper statistical model/test (parametric or non-parametric) after ddct method; and I still have not seen it in the text. A one-way ANOVA or t-test would be enough for you. Once the statistical metrics have not been properly addressed I can not trust any qPCR results.

I strongly suggest you to perform the statistical test and potential corrections and resubmit the manuscript.

Responses from the authors: We are very sorry that our misunderstanding did not effectively solve the problems raised by the reviewer, which increased your workload. Your patient explanation gives us a correct understanding.Thanks for your excellent summary comments. Your points are well taken. Per your suggestion,We have modified and supplemented this part.

After comparing the genes expression levels of different groups with 2-△△Ct method.The data from all experiments were first tested for collation and outliers. Statistical analysis was performed using SPSS software, version 20.0 (IBM-SPSS, Chicago, Illinois, USA).One-way ANOVA and Duncan multiple comparisons test were used to evaluate the differences expression of different genes in different varieties. The data were presented as mean±standard deviation (SD). 

We have carefully conducted statistical tests and potential corrections.We are looking forward to your "acceptance" suggestions and responses！ If there is anything else we need to do, please feel free to let us know.

---

## [Editor Report · Decision Letter 3]

16 Apr 2023

Fatty acid binding protein 1 and fatty acid synthetase over-expression have differential effects on collagen III synthesis and cross-linking in Zongdihua pig primary adipocytes

PONE-D-22-15349R3

Dear Dr. Zhou,

We’re pleased to inform you that your manuscript has been judged scientifically suitable for publication and will be formally accepted for publication once it meets all outstanding technical requirements.

Kind regards,

Marcio Duarte, PhD

Academic Editor

PLOS ONE

Additional Editor Comments (optional):

All comments made in the last revision were adequately addressed. The manuscript is ready for publication.
---

## [Editor Report · Acceptance letter]

25 Apr 2023

PONE-D-22-15349R3 

Fatty acid binding protein 1 and fatty acid synthetase over-expression have differential effects on collagen III synthesis and cross-linking in Zongdihua pig primary adipocytes 

Dear Dr. Zhou:

I'm pleased to inform you that your manuscript has been deemed suitable for publication in PLOS ONE. Congratulations! Your manuscript is now with our production department. 

Kind regards, 

on behalf of

Dr. Marcio Duarte 

Academic Editor

PLOS ONE